# Markov Persuasion Processes: Learning to Persuade From Scratch

**Francesco Bacchiocchi**[*]
Politecnico di Milano
francesco.bacchiocchi@polimi.it

**Francesco Emanuele Stradi**[*]
Politecnico di Milano
francescoemanuele.stradi@polimi.it

**Matteo Castiglioni**
Politecnico di Milano
matteo.castiglioni@polimi.it

**Alberto Marchesi**
Politecnico di Milano
alberto.marchesi@polimi.it

**Nicola Gatti**
Politecnico di Milano
nicola.gatti@polimi.it

## Abstract

In *Bayesian persuasion*, an informed sender strategically discloses information to a receiver so as to persuade them to undertake desirable actions. Recently, *Markov persuasion processes* (MPPs) have been introduced to capture *sequential* scenarios where a sender faces a stream of myopic receivers in a Markovian environment. The MPPs studied so far in the literature suffer from issues that prevent them from being fully operational in practice, *e.g.*, they assume that the *sender knows receivers' rewards*. We fix such issues by addressing MPPs where the sender has no knowledge about the environment. We design a learning algorithm for the sender, working with partial feedback. We prove that its regret with respect to an optimal information-disclosure policy grows sublinearly in the number of episodes, as it is the case for the loss in persuasiveness cumulated while learning. Moreover, we provide lower bounds for our setting matching the guarantees of our algorithm.

## 1 Introduction

*Bayesian persuasion* [Kamenica and Gentzkow, 2011] studies how an informed sender should strategically disclose information to influence the behavior of a self-interested receiver. Bayesian persuasion has received a growing attention over the last years, since it captures several fundamental problems arising in real-world applications, such as, *e.g.*, online advertising [Bro Miltersen and Sheffet, 2012, Emek et al., 2014, Badanidiyuru et al., 2018, Bacchiocchi et al., 2022], voting [Cheng et al., 2015, Alonso and Câmara, 2016, Castiglioni et al., 2020a, Castiglioni and Gatti, 2021], traffic routing [Vasserman et al., 2015, Bhaskar et al., 2016, Castiglioni et al., 2021a], recommendation systems [Mansour et al., 2016], e-commerce [Castiglioni et al., 2022], security [Rabinovich et al., 2015, Xu et al., 2016], marketing [Babichenko and Barman, 2017, Candogan, 2019], clinical trials [Kolotilin, 2015], and financial regulation [Goldstein and Leitner, 2018].

Most of the works on Bayesian persuasion focus on *one-shot* interactions, where information disclosure is performed "one shot". Despite real-world problems are usually *sequential*, there are only few exceptions that consider multi-step information disclosure [Wu et al., 2022, Gan et al., 2022,

---

[*]Equal contribution.

39th Conference on Neural Information Processing Systems (NeurIPS 2025).

2023, Bernasconi et al., 2022, 2023b, Iyer et al., 2023, Lin et al., 2024]. Specifically, Wu et al. [2022] initiated the study of *Markov persuasion processes* (MPPs), which model scenarios where a sender sequentially faces a stream of *myopic* receivers in an unknown Markovian environment. In each state of the environment, the sender privately observes some information—encoded in an outcome stochastically determined according to a prior distribution—and faces a *new* receiver, who is then called to take an action. The outcome and the receiver's action jointly determine the agents' rewards and the next state. In an MPP, the goal of the sender goal is to disclose information about the outcome at each state, so as to persuade the receivers to take actions maximizing sender's *long-term* rewards.

The MPP formalism finds application in several real-world settings, such as e-commerce and recommendation systems [Wu et al., 2022]. For example, an MPP can model the problem faced by an online streaming platform recommending movies to its users. The platform has an informational advantage over users (*e.g.*, it has access to views statistics), and it exploits available information to induce users to watch suggested movies, so as to maximize views. The MPPs studied by Wu et al. [2022] rely on some limiting assumptions that prevent them from being fully operational in practice. For instance, they make the assumption that the *sender has perfect knowledge of receiver's rewards*. In the online streaming platform example, this assumption requires that the platform knows everything about users' (private) preferences over movies, which is unrealistic in practice.

## 1.1 Original Contributions

We relax the assumptions of Wu et al. [2022], by addressing MPPs where *the sender does not know anything about the environment*. We consider settings in which they have no knowledge about transitions, prior distributions over outcomes, sender's stochastic rewards, and receivers' ones. Ideally, the goal is to design learning algorithms that are *persuasive* and attain *regret* sublinear in the number of episodes of learning $T$. The regret is the difference between sender's rewards cumulated over the episodes and what would have been obtained by always using an optimal information-disclosure policy. Persuasiveness is about ensuring receivers are incentivized to take desired actions. Learning in MPPs *without knowledge of receivers' rewards* begets considerable additional challenges compared to the case of Wu et al. [2022]. Indeed, the latter design a sublinear-regret algorithm that is persuasive at every episode with high probability, while we show that this is *not* attainable in our setting. Intuitively, this is due to the fact that, since the sender does *not* know receivers' rewards, some episodes must be used to learn how to be "approximately" persuasive. As a consequence, in this work we look for algorithms that attain sublinear regret while ensuring that the cumulative *violation* of persuasiveness grows sublinearly in $T$. This is the most natural requirement in all the cases where persuasiveness cannot be achieved at every episode, and it has already been addressed in related settings (see, *e.g.*, [Bernasconi et al., 2022, Cacciamani et al., 2023, Gan et al., 2023]).

As a warm-up, we start studying a *full* feedback case where, after each episode, the sender observes the reward associated with every possible action in all the state-outcome pairs encountered during the episode. We propose an algorithm, called Optimistic Persuasive Policy Search (OPPS), which uses information-disclosure policies computed by being *optimistic* with respect to both sender's expected rewards and persuasiveness requirements. We show that, under full feedback, OPPS attains $\widetilde{\mathcal{O}}(\sqrt{T})$ regret and violation. Then, we switch to the *partial* feedback case, where the sender only observes the rewards for the state-outcome-action triplets actually visited during the episode. We extend the OPPS algorithm to this setting, by adding a preliminary *exploration* phase having the goal of gathering as much feedback as possible about persuasiveness. After that, the algorithm switches to an optimistic approach over information-disclosure policies that are "approximately" persuasive. We prove that OPPS with partial feedback attains $\widetilde{\mathcal{O}}(T^\alpha)$ regret and $\widetilde{\mathcal{O}}(T^{1-\alpha/2})$ violation, where $\alpha \in [1/2, 1]$ is a parameter controlling the amount of exploration. Finally, we provide a lower bound showing that the trade-off between regret and violation achieved by means of OPPS is tight.

## 1.2 Related Works

We refer to Appendix A for additional related works.

The work most related to ours is [Wu et al., 2022], studying MPPs where the sender knows everything about receivers' rewards, with the only elements unknown to them being their rewards, transition probabilities, and prior distributions. Moreover, Wu et al. [2022] also assume that the receivers know everything about the environment, so as to select a best-response action, and that all rewards are

deterministic. In contrast, we consider MPPs in which sender and receivers have no knowledge of the environment, including their rewards, which we assume to be stochastic. Other related works are [Gan et al., 2022], studying Bayesian persuasion problems where a sender sequentially interacts with a myopic receiver in a multi-state environment, and [Bernasconi et al., 2023b], addressing MPPs with a farsighted receiver. These two works considerably depart from ours, as they both assume that the sender knows everything about the environment, including transitions, priors, and rewards. Thus, they are *not* concerned with learning problems. Finally, [Bernasconi et al., 2022] studies settings where a sender faces a farsighted receiver in a sequential environment with a tree structure, addressing the case in which the only elements unknown to the sender are the prior distributions over outcomes, while rewards are deterministic and known. The tree structure considerably eases learning, as it intuitively allows to factor the uncertainty about transitions in the rewards at the leaves of the tree.

Our work is also related to learning in one-shot Bayesian persuasion played repeatedly [Castiglioni et al., 2020b, 2021b, Zu et al., 2021, Bernasconi et al., 2023a], and works on online *Markov decision processes* (MDPs) [Auer et al., 2008, Even-Dar et al., 2009, Neu et al., 2010, Rosenberg and Mansour, 2019, Jin et al., 2020], in particular those on constrained MDPs [Wei et al., 2018, Zheng and Ratliff, 2020, Efroni et al., 2020, Qiu et al., 2020, Stradi et al., 2024].

## 2 Preliminaries

### 2.1 Bayesian Persuasion

The classical *Bayesian persuasion* framework introduced by Kamenica and Gentzkow [2011] models a *one-shot* interaction between a *sender* and a *receiver*. The latter has to take an action $a$ from a finite set $A$, while the former privately observes an outcome $\omega$ sampled from a finite set $\Omega$ according to a prior distribution $\mu \in \Delta(\Omega)$, which is *known to both* the sender and the receiver.[2] The rewards of both agents depend on the receiver's action and the realized outcome, as defined by the functions $r_S, r_R : \Omega \times A \to [0, 1]$, where $r_R(\omega, a)$ and $r_S(\omega, a)$ denote the rewards of the sender and the receiver, respectively, when the outcome is $\omega \in \Omega$ and action $a \in A$ is played. The sender can strategically disclose information about the outcome to the receiver, by *publicly committing to* a signaling scheme $\phi$, which is a randomized mapping from outcomes to signals being sent to the receiver. Formally, $\phi : \Omega \to \Delta(\mathcal{S})$, where $\mathcal{S}$ denotes a suitable finite set of signals. For ease of notation, we let $\phi(\cdot|\omega) \in \Delta(\mathcal{S})$ be the probability distribution over signals employed by the sender when the realized outcome is $\omega \in \Omega$, with $\phi(s|\omega)$ being the probability of sending signal $s \in \mathcal{S}$.

The sender-receiver interaction goes as follows: (i) the sender publicly commits to a signaling scheme $\phi$; (ii) the sender observes the realized outcome $\omega \sim \mu$ and draws a signal $s \sim \phi(\cdot|\omega)$; and (iii) the receiver observes the signal $s$ and plays an action. Specifically, after observing $s$ under a signaling scheme $\phi$, the receiver infers a *posterior* distribution over outcomes and plays a *best-response* action $b^{\phi}(s) \in A$ according to such distribution. Formally, $b^{\phi}(s) \in \arg\max_{a \in A} \sum_{\omega \in \Omega} \mu(\omega)\phi(s|\omega)r_R(\omega, a)$, where the expression being maximized encodes the (unnormalized) expected reward of the receiver. As it is customary in the literature (see, *e.g.*, [Dughmi and Xu, 2016]), we assume that the receiver breaks ties in favor of the sender, by selecting a best response maximizing sender's expected reward when multiple best responses are available.

The goal of the sender is to commit to a signaling scheme $\phi$ that maximizes their expected reward, which is computed as follows: $\sum_{\omega \in \Omega} \mu(\omega) \sum_{s \in \mathcal{S}} \phi(s|\omega)r_S(\omega, b^{\phi}(s))$.

### 2.2 Markov Persuasion Processes

An MPP [Wu et al., 2022] generalizes one-shot Bayesian persuasion to settings where the sender faces a stream of receivers in an MDP, with each receiver *myopically* taking an action maximizing immediate reward. An (*episodic*) MPP is a tuple $M := (X, A, \Omega, \mu, P, \{r_{S,t}\}_{t=1}^{T}, \{r_{R,t}\}_{t=1}^{T})$:

- $T$ is the number of episodes.[3]
- $X$, $A$, and $\Omega$ are finite sets of states, actions, and outcomes, respectively.
- $\mu : X \to \Delta(\Omega)$ is a prior function defining a probability distribution over outcomes at each state. We let $\mu(\omega|x)$ be the probability of sampling outcome $\omega \in \Omega$ in state $x \in X$.

---

[2]In this work, we denote by $\Delta(X)$ the set of all the probability distributions having set $X$ as support.

[3]We denote an episode by $t \in [T]$, where $[a \ldots b]$ is the set of all integers from $a$ to $b$ and $[b] := [1 \ldots b]$.

- $P : X \times \Omega \times A \to \Delta(X)$ is a transition function. We let $P(x'|x,\omega,a)$ be the probability of going from $x \in X$ to $x' \in X$ by taking action $a \in A$, when the outcome in state $x$ is $\omega \in \Omega$.
- $\{r_{S,t}\}_{t=1}^T$ is a sequence specifying a sender's reward function $r_{S,t} : X \times \Omega \times A \to [0,1]$ at each episode $t$. Given $x \in X$, $\omega \in \Omega$, and $a \in A$, each $r_{S,t}(x,\omega,a)$ for $t \in [T]$ is sampled independently from a distribution $\nu_S(x,\omega,a) \in \Delta([0,1])$ with mean $r_S(x,\omega,a)$.
- $\{r_{R,t}\}_{t=1}^T$ is a sequence defining a receivers' reward function $r_{R,t} : X \times \Omega \times A \to [0,1]$ at each episode $t$. Given $x \in X$, $\omega \in \Omega$, and $a \in A$, each $r_{R,t}(x,\omega,a)$ for $t \in [T]$ is sampled independently from a distribution $\nu_R(x,\omega,a) \in \Delta([0,1])$ with mean $r_R(x,\omega,a)$.

We focus w.l.o.g. on *loop-free* episodic MPPs, as customary in online learning in MDPs (see, *e.g.*, [Rosenberg and Mansour, 2019]). In a loop-free MPP, states are partitioned into $L+1$ layers $X_0, \ldots, X_L$ such that $X_0 := \{x_0\}$ and $X_L := \{x_L\}$, with $x_0$ being the initial state starting the episode and $x_L$ being the final one, in which the episode ends. Moreover, by letting $\mathcal{K} := [0 \ldots L-1]$ for ease of notation, $P(x'|x,\omega,a) > 0$ only when $x' \in X_{k+1}$ and $x \in X_k$ for some $k \in \mathcal{K}$.[4]

At each episode of an episodic MPP, the sender commits to a *signaling policy* $\phi : X \times \Omega \to \Delta(\mathcal{S})$, which defines a probability distribution over a finite set $\mathcal{S}$ of signals for the receivers for every state $x \in X$ and outcome $\omega \in \Omega$. For ease of notation, we denote by $\phi(\cdot|x,\omega) \in \Delta(\mathcal{S})$ such probability distributions, with $\phi(s|x,\omega)$ being the probability of sending a signal $s \in \mathcal{S}$ in state $x$ when the realized outcome is $\omega$. Similarly to one-shot Bayesian persuasion, a myopic receiver acting at state $x \in X$ and receiving signal $s \in \mathcal{S}$ infers a posterior distribution over outcomes and plays a best-response action. We denote by $b^\phi(s,x) \in A$ the best response played by such a receiver under the signaling policy $\phi$ (assuming ties are broken in favor of the sender).

As customary in Bayesian persuasion (see, *e.g.*, [Arieli and Babichenko, 2019]), a revelation-principle-style argument allows to focus w.l.o.g. on signaling policies that are direct and persuasive. Formally, a signaling policy is *direct* if the set of signals coincides with the set of actions, namely $\mathcal{S} = A$. Intuitively, signals should be interpreted as action recommendations for the receivers. Moreover, a direct signaling policy is said to be *persuasive* if it incentivizes the receivers to follow recommendations. Formally, $\phi : X \times \Omega \to \Delta(A)$ is persuasive if for every state $x \in X$ and recommendation $a \in A$,

$$\sum_{\omega \in \Omega} \mu(\omega|x)\phi(a|x,\omega)\big(r_R(x,\omega,a) - r_R(x,\omega,b^\phi(a,x))\big) \geq 0.$$

Intuitively, the inequality above states that a receiver acting at state $x$ is better off following sender's recommendation to play action $a$, since by doing so they get an (unnormalized) expected reward greater than or equal to what they would obtain by playing a best-response action $b^\phi(a,x)$.

Algorithm 1 shows the interaction between sender and receivers at $t \in [T]$. Sender and receivers do *not* know anything about the transition function $P$, the prior function $\mu$, and the rewards $r_{S,t}(x,\omega,a), r_{R,t}(x,\omega,a)$ (including their distributions). At the end of each episode, the sender gets to know the triplets $(x_k, \omega_k, a_k)$—for all $k \in \mathcal{K}$—that are *visited* during the episode, and an additional *feedback* about rewards. In this work, we consider two types of feedback. The first one—called *full* feedback—encompasses all agents' rewards for the pairs $(x_k, \omega_k)$ visited during the episode, *i.e.*, the rewards for all the triplets $(x_k, \omega_k, a)$ for $a \in A$. The second

> **Algorithm 1** Sender-Receivers Interaction at $t \in [T]$
>
> 1: The rewards $r_{S,t}(x,\omega,a), r_{R,t}(x,\omega,a)$ are sampled
> 2: Sender publicly commits to $\phi_t : X \times \Omega \to \Delta(A)$
> 3: The state of the MPP is initialized to $x_0$
> 4: **for** $k = 0, \ldots, L-1$ **do**
> 5:     Sender observes outcome $\omega_k \sim \mu(x_k)$
> 6:     Sender draws recommendation $a_k \sim \phi(\cdot|x_k,\omega_k)$
> 7:     A *new* Receiver observes $a_k$ and plays it
> 8:     The MPP evolves to $x_{k+1} \sim P(\cdot|x_k,\omega_k,a_k)$
> 9:     Sender observes the next state $x_{k+1}$
> 10: **end for**
> 11: Sender observes *feedback* for every $k \in [0 \ldots L-1]$:
>     • *full* $\to [r_{S,t}(x_k,\omega_k,a), r_{R,t}(x_k,\omega_k,a)]_{a \in A}$
>     • *partial* $\to r_{S,t}(x_k,\omega_k,a_k), r_{R,t}(x_k,\omega_k,a_k)$

type—called *partial* feedback—only consists in agents' rewards for the visited triplets $(x_k, \omega_k, a_k)$.[5]

Notice that Algorithm 1 assumes that receivers always play recommended actions. This is standard in settings where the sender has *not* enough information to be persuasive at every episode, and

---

[4]The loop-free property is w.l.o.g. since any episodic MPP with finite horizon $H$ that is *not* loop-free can be cast into a loop-free one by duplicating states $H$ times, *i.e.*, $x \in X$ is mapped to new states $(x,k)$ with $k \in [H]$.

[5]In this work we use the adjective *full* to refer to a type of feedback that is *not* the most informative one. Indeed, a full feedback according to the classical terminology used in online learning [Cesa-Bianchi and Lugosi, 2006, Orabona, 2019] would encompass agents' rewards for all the possible triplets $(x,\omega,a)$, while full feedback in our terminology only consists in the rewards for the triplets with $x = x_k$ and $\omega = \omega_k$ for some $k \in \mathcal{K}$.

it motivates why learning algorithms are designed to guarantee that the cumulative violation of persuasiveness grows sublinearly in $T$, or, equivalently, that the per-round violation of persuasiveness goes to zero as $T$ grows [Bernasconi et al., 2022, Cacciamani et al., 2023, Gan et al., 2023]. Indeed, this ensures that it is in the receivers' best interest to stick to recommendations.

## 3 The Learning Problem

In this section, we formally introduce the learning problem tackled in the rest of the paper. First, in Section 3.1, we extend the notion of occupancy measure to MPPs. In Section 3.2, we formally introduce learning objectives. Finally, in Section 3.3, we provide some preliminary elements needed by our algorithms, developed in Sections 4 and 5. The proofs of all our results are in Appendixes D and E.

### 3.1 Occupancy Measures in MPPs

Next, we extend the well-known notion of *occupancy measure* of an MDP [Rosenberg and Mansour, 2019] to MPPs. Given a transition function $P$, a signaling policy $\phi$, and a prior function $\mu$, the occupancy measure induced by $P$, $\phi$, and $\mu$ is a vector $q^{P,\phi,\mu} \in [0,1]^{|X \times \Omega \times A \times X|}$ whose entries are specified as follows. For every $x \in X_k, \omega \in \Omega, a \in A$, and $x' \in X_{k+1}$ with $k \in \mathcal{K}$, it holds:

$$q^{P,\phi,\mu}(x,\omega,a,x') := \mathbb{P}\Big\{ (x_k, \omega_k, a_k, x_{k+1}) = (x, \omega, a, x') \mid P, \phi, \mu \Big\},$$

which is the probability that the next state is $x'$ after the receiver plays action $a$ in state $x$ when the realized outcome is $\omega$, under transition function $P$, signaling policy $\phi$, and prior $\mu$. Moreover, we also let $q^{P,\phi,\mu}(x,\omega,a) := \sum_{x' \in X_{k+1}} q^{P,\phi,\mu}(x,\omega,a,x')$, $q^{P,\phi,\mu}(x,\omega) := \sum_{a \in A} q^{P,\phi,\mu}(x,\omega,a)$, and $q^{P,\phi,\mu}(x) := \sum_{\omega \in \Omega} q^{P,\phi,\mu}(x,\omega)$, for the ease of notation.

The following lemma characterizes the set of *valid* occupancy measures and it is a generalization to the MPP setting of a similar lemma by Rosenberg and Mansour [2019].

**Lemma 1.** *A vector $q \in [0,1]^{|X \times \Omega \times A \times X|}$ is a valid occupancy measure of an MPP if and only if*

$$(i) \quad \sum_{x \in X_k} \sum_{\omega \in \Omega} \sum_{a \in A} \sum_{x' \in X_{k+1}} q(x,\omega,a,x') = 1 \quad \forall k \in \mathcal{K},$$

$$(ii) \quad \sum_{x' \in X_{k-1}} \sum_{\omega \in \Omega} \sum_{a \in A} q(x',\omega,a,x) = q(x) \quad \forall k \in [1 \ldots L-1], \forall x \in X_k,$$

$$(iii) \quad P^q = P \text{ and } \mu^q = \mu,$$

*where $P$ is the transition function of the MPP and $\mu$ its prior function, while $P^q$ and $\mu^q$ are the transition and prior functions, respectively, induced by $q$ (see definitions below).*

As it is the case in standard MDPs, a valid occupancy measure $q \in [0,1]^{|X \times \Omega \times A \times X|}$ induces a transition function $P^q$ and a signaling policy $\phi^q$. Moreover, in an MPP, a valid occupancy measure also induces a prior function $\mu^q$. These are defined as follows:

$$P^q(x'|x,\omega,a) := \frac{q(x,\omega,a,x')}{q(x,\omega,a)}, \ \phi^q(a|x,\omega) := \frac{q(x,\omega,a)}{q(x,\omega)}, \ \mu^q(\omega|x) := \frac{q(x,\omega)}{q(x)}.$$

Thus, using valid occupancy measures is *equivalent* to using signaling policies. In the following, we denote by $\mathcal{Q} \subseteq [0,1]^{|X \times \Omega \times A \times X|}$ the set of all the valid occupancy measures of an MPP.

### 3.2 Learning Objectives

Our goal is to design learning algorithms for the sender in an episodic MPP. We would like algorithms that prescribe sequences of signaling policies $\phi_t$ that maximize sender's cumulative reward over the $T$ episodes, while at the same time guaranteeing that the violation of persuasiveness constraints is bounded. Notice that, differently from Wu et al. [2022], we do *not* aim at designing learning algorithms whose policies $\phi_t$ are persuasive at every episode $t$ with high probability, since this is unattainable in our setting in which the sender does *not* know anything about the environment (see Theorem 6). Thus, in this paper we pursue a different objective, formally described in the following.

**Baseline**   First, we introduce the baseline used to evaluate sender's performances. This is defined as the value of the optimization problem faced by the sender in the *offline* version of the MPP. Such a problem is concerned with expectations of the stochastic quantities in the episodic MPP. By exploiting occupancy measures, the problem can be formulated as the following linear program:

$$\max_{q \in \mathcal{Q}} \quad \sum_{x \in X} \sum_{\omega \in \Omega} \sum_{a \in A} q(x, \omega, a) r_S(x, \omega, a) \quad \text{s.t.} \tag{1a}$$

$$\sum_{\omega \in \Omega} q(x, \omega, a) \Big( r_R(x, \omega, a) - r_R(x, \omega, a') \Big) \geq 0 \quad \forall x \in X, \forall \omega \in \Omega, \forall a \in A, \forall a' \neq a \in A. \tag{1b}$$

Intuitively, Problem (1) computes an occupancy measure (or, equivalently, signaling policy) maximizing sender's expected reward subject to persuasiveness constraints. By letting $r_S \in [0, 1]^{|X \times \Omega \times A|}$ be the vector whose entries are the mean values $r_S(x, \omega, a)$ of sender's rewards, our baseline is defined as $\text{OPT} := r_S^\top q^*$, where $q^* \in \mathcal{Q}$ denotes an optimal solution to Problem (1). In the following, we denote by $\phi^*$ an optimal signaling policy, defined as $\phi^* := \phi^{q^*}$.

**Metrics**   We evaluate the performances of learning algorithms by means of two distinct metrics. The first one is the *(cumulative) regret* $R_T$, which accounts for the difference between the cumulative sender's expected reward obtained by always playing $\phi^*$ and that achieved by using the signaling policies $\phi_t$ prescribed by the algorithm. Formally,

$$R_T := T \cdot \text{OPT} - \sum_{t \in [T]} r_S^\top q_t = \sum_{t \in [T]} r_S^\top \big( q^* - q_t \big),$$

where we let $q_t := q^{P, \phi_t, \mu}$ be the occupancy measure induced by $\phi_t$. The second metric used to evaluate learning algorithms is the *(cumulative) violation* $V_T$, which is formally defined as follows:

$$V_T := \sum_{t \in [T]} \sum_{x \in X} \sum_{\omega \in \Omega} \sum_{a \in A} q_t(x, \omega, a) \cdot \big( r_R(x, \omega, b^\phi(a, x)) - r_R(x, \omega, a) \big).$$

Intuitively, $V_T$ encodes the overall expected loss in persuasiveness over the $T$ episodes. Our goal is to develop learning algorithms that prescribe signaling policies $\phi_t$ guaranteeing that both $R_T$ and $V_T$ grow sublinearly in $T$, namely $R_T = o(T)$ and $V_T = o(T)$.

### 3.3   Estimators and Confidence Bounds

Before delving in algorithm design, we introduce estimators and confidence bounds for the stochastic quantities involved in an MPP, namely, transitions, priors, sender's rewards, and receivers' ones. In the following, we let $N_t(x, \omega, a, x') \in \mathbb{N}$ be the number of episodes up to $t \in [T]$ (this excluded) in which the tuple $(x, \omega, a, x')$ is visited. Formally, $N_t(x, \omega, a, x') := \sum_{\tau \in [t-1]} \mathbb{1}_\tau \{x, \omega, a, x'\}$, where the indicator function is 1 if and only if the tuple is visited at $\tau$. Similarly, we define the counters $N_t(x, \omega, a, ), N_t(x, \omega),$ and $N_t(x)$ in terms of their respective indicators $\mathbb{1}_\tau \{x, \omega, a\}, \mathbb{1}_\tau \{x, \omega\},$ and $\mathbb{1}_\tau \{x\}$, which are equal to 1 if and only if $(x, \omega, a), (x, \omega),$ and $x$, respectively, are visited at $\tau$.

Next, we define the estimators employed by our algorithms. At the beginning of each episode $t \in [T]$, the estimated probability of going from $x \in X$ to $x' \in X$ by playing $a \in A$, when the outcome realized in state $x$ is $\omega \in \Omega$, is equal to $\overline{P}_t(x'|x, \omega, a) := \frac{N_t(x, \omega, a, x')}{\max\{1, N_t(x, \omega, a)\}}$. Moreover, for every $x \in X$ and $\omega \in \Omega$, the estimated probability of sampling outcome $\omega$ from the prior probability distribution at state $x$ is defined as $\overline{\mu}_t(\omega|x) := \frac{N_t(x, \omega)}{\max\{1, N_t(x)\}}$. Finally, for every state $x \in X$, outcome $\omega \in \Omega$, and action $a \in A$, the estimated sender's and receivers' rewards are defined as $\overline{r}_{S,t}(x, \omega, a) := \frac{\sum_{\tau \in [t-1]} r_{S,\tau}(x, \omega, a) \mathbb{1}_\tau \{x, \omega, a\}}{\max\{1, N_t(x, \omega, a)\}}$, and $\overline{r}_{R,t}(x, \omega, a) := \frac{\sum_{\tau \in [t-1]} r_{R,\tau}(x, \omega, a) \mathbb{1}_\tau \{x, \omega, a\}}{\max\{1, N_t(x, \omega, a)\}}$.

Appendix B formally defines the confidence bounds employed by our algorithms. For the transition function $P$, at each $t \in [T]$, for every $x \in X$, $\omega \in \Omega$, and $a \in A$, we provide a confidence bound $\epsilon_t(x, \omega, a)$ for the probability distribution over next states associated with the triplet $(x, \omega, a)$, where the distance between distributions is expressed in $\|\cdot\|_1$-norm (see Lemma 4). Similarly, we provide a confidence bound $\zeta_t(x)$ in terms of $\|\cdot\|_1$-norm for the prior $\mu(x)$ at each state $x \in X$ (see Lemma 5). Moreover, for every $x \in X$, $\omega \in \Omega$, and $a \in A$, we provide confidence bounds $\xi_{S,t}(x, \omega, a)$ and $\xi_{R,t}(x, \omega, a)$ for sender's and receivers' rewards, respectively, associated with the triplet $(x, \omega, a)$

(see Lemmas 6 and 7 for the full feedback case, while Lemmas 8 and 9 for the partial feedback one). In conclusion, for ease of presentation, for a confidence parameter $\delta \in (0, 1)$, we refer to the event in which all the confidence bounds hold—called *clean event*—as $\mathcal{E}(\delta)$. By combining all the lemmas in Appendix B, $\mathcal{E}(\delta)$ holds with probability at least $1 - 4\delta$ (by applying a union bound).

# 4 The Full Feedback Case

We first address settings with full feedback, as a warm-up towards the analysis of partial feedback.

## 4.1 The OPPS Algorithm With Full Feedback

We propose an algorithm called Optimisitc Persuasive Policy Search (OPPS). At each episode, the algorithm solves a variation of the offline optimization problem (Problem (1)), called Opt-Opt, obtained by substituting mean values with upper/lower confidence bounds. Specifically, Opt-Opt is *optimistic* with respect to *both* sender's rewards and persuasiveness constraints satisfaction. For reasons of space, we defer Opt-Opt to Problem (2) in Appendix C. Crucially, by using occupancy measures, Opt-Opt can be formulated as an LP, and, thus, solved efficiently. Notice that, since confidence bounds for $P$ and $\mu$ are expressed in terms of $||\cdot||_1$-norm, in order to formulate Opt-Opt as an LP we need some additional variables and linear constraints, as described in detail in Appendix C.

Algorithm 2 provides the pseudocode of OPPS with *full* feedback. At each $t \in [T]$, the algorithm first updates all the estimators and confidence bounds according to the feedback received in previous episodes (Line 3). Then, it commits to the signaling policy $\phi_t$ induced by an optimal solution $\widehat{q}_t$ to Opt-Opt, computed in Line 4. Notice that, the occupancy measure $q_t$ resulting from committing to $\phi_t$ (and used in the definitions of $R_T$ and $V_T$) is in general different from $\widehat{q}_t$, as the former is defined in

---

**Algorithm 2** Optimistic Persuasive Policy Search *(full)*

**Require:** $X, A, T$, confidence parameter $\delta \in (0, 1)$
1: Initialize all estimators to 0 and all bounds to $+\infty$
2: **for** $t = 1, \ldots, T$ **do**
3:     Update all estimators $\overline{P}_t, \overline{\mu}_t, \overline{r}_{S,t}, \overline{r}_{R,t}$ and bounds $\epsilon_t, \zeta_t, \xi_{S,t}, \xi_{R,t}$ given new observations
4:     $\widehat{q}_t \leftarrow$ Solve Opt-Opt (Problem (2))
5:     $\phi_t \leftarrow \phi^{\widehat{q}_t}$
6:     Run Algorithm 1 by committing to $\phi_t$
7:     Observe *full* feedback from Algorithm 1
8: **end for**

---

terms of the true (and unknown) transition and prior functions, namely $P$ and $\mu$.

## 4.2 Algorithm Analysis With Full Feedback

Next, we prove the guarantees of OPPS with *full* feedback. The first crucial component is the following lemma, which shows that Opt-Opt admits a feasible solution at every episode with high probability.

**Lemma 2.** *Given $\delta \in (0, 1)$, under event $\mathcal{E}(\delta)$, Opt-Opt admits a feasible solution at every $t \in [T]$.*

Intuitively, Lemma 2 is proved by showing (a) that Problem 1 always admits a feasible solution, which is the occupancy measure $q^\diamond$ induced by the signaling policy that fully reveals outcomes to the receiver, and (b) that $q^\diamond$ is a feasible solution to Opt-Opt at every episode, under $\mathcal{E}(\delta)$. Point (b) holds thanks to the fact that Opt-Opt optimistically accounts for persuasiveness constraints satisfaction, by using suitable upper and lower confidence bounds.

The second crucial component of our analysis is a relation between the occupancy measures $\widehat{q}_t$ computed by the OPPS algorithm and the occupancy measures $q_t$ that actually result from committing to $\phi_t$ under the true transitions and priors. This is formally stated by the following lemma.

**Lemma 3.** *Given any $\delta \in (0, 1)$, under the clean event $\mathcal{E}(\delta)$, with probability at least $1 - 2\delta$, it holds that $\sum_{t \in [T]} \|q_t - \widehat{q}_t\|_1 \le \mathcal{O}\left(L^2 |X| \sqrt{T |A| |\Omega| \ln(T |X| |\Omega| |A| / \delta)}\right).$*

Intuitively, Lemma 3 is proved by an inductive argument that relates the uncertainty associated with both the transition and the prior functions to the $\|\cdot\|_1$-norm difference between $q_t$ and $\widehat{q}_t$ cumulated over the episodes. Lemmas 2 and 3 pave the way to our two main theorems for the full feedback setting. The first theorem bounds the regret $R_T$ achieved by OPPS, while the second one bounds its cumulative violation $V_T$. Formally:

**Theorem 1.** *Given any $\delta \in (0,1)$, with probability at least $1 - 7\delta$, Algorithm 2 attains regret* $R_T \leq \widetilde{\mathcal{O}}\left(L^2|X|\sqrt{T|A||\Omega|\ln(1/\delta)}\right).$

**Theorem 2.** *Given $\delta \in (0,1)$, with probability at least $1 - 7\delta$, Algorithm 2 attains violation* $V_T \leq \widetilde{\mathcal{O}}\left(L^2|X|\sqrt{T|A||\Omega|\ln(1/\delta)}\right).$

In conclusion, in the *full* feedback case, OPPS attains $R_T$ and $V_T$ growing as $\widetilde{\mathcal{O}}(\sqrt{T})$. Intuitively, this is made effective by the fact that all the estimators concentrate at a $1/\sqrt{T}$ rate. As shown next, achieving such regret and violation bounds is *not* possible anymore under *partial* feedback.

### 4.3 Lower Bound

We conclude the section by showing that the regret and violation bounds attained by the OPPS algorithm with *full* feedback are tight. Formally:

**Theorem 3.** *Let $\delta \in (0, 1/4)$. For every algorithm that guarantees $R_T \leq \mathcal{O}(\sqrt{T})$ with probability at least $1 - \delta$, there exists an absolute constant $\psi > 0$ and a problem instance in which the algorithm must have $V_T \geq \Omega(\sqrt{T})$ with probability larger than $\psi$.*

## 5 The Partial Feedback Case

In this section, we switch the attention to partial feedback. The crucial aspect that makes the case of partial feedback more challenging than the one of full feedback is that, after committing to a signaling policy $\phi_t$, the sender does *not* observe sufficient feedback about the persuasiveness of $\phi_t$. This makes achieving sublinear violation in the partial feedback case much harder than in the full feedback case. In order to overcome such a challenge, some episodes of learning must be devoted to the estimation of the quantities involved in persuasiveness constraints. This is necessary to build a suitable approximation of such constraints to be exploited in the remaining episodes, in which an optimistic approach similar to that employed with full feedback

---

**Algorithm 3** Optimistic Persuasive Policy Search *(partial)*

**Require:** $X, \Omega, A, T, \delta \in (0,1), \alpha \in [0,1]$
1: $N \leftarrow \lceil T^\alpha \rceil$
2: Initialize all estimators to 0 and all bounds to $+\infty$
3: Initialize counter $C(x, \omega, a)$ to 0 for all $(x, \omega, a)$
4: **for** $t = 1, \ldots, T$ **do**
5:     Update all estimators $\overline{P}_t, \overline{\mu}_t, \overline{r}_{S,t}, \overline{r}_{R,t}$ and bounds $\epsilon_t, \zeta_t, \xi_{S,t}, \xi_{R,t}$ given new observations
6:     **if** $t \leq N|X||\Omega||A|$ **then**
7:         $(x, \omega, a) \leftarrow \arg\min_{(x,\omega,a) \in X \times \Omega \times A} C(x, \omega, a)$
8:         $\widehat{q}_t \leftarrow$ Solve Opt-Opt with its objective modified as $\sum_{x' \in X} q(x, \omega, a, x')$
9:         $C(x, \omega, a) \leftarrow C(x, \omega, a) + 1$
10:    **else**
11:        $\widehat{q}_t \leftarrow$ Solve Opt-Opt (Problem (2), Appendix C)
12:    **end if**
13:    $\phi_t \leftarrow \phi^{\widehat{q}_t}$
14:    Run Algorithm 1 by committing to $\phi_t$
15:    Observe *partial* feedback from Algorithm 1
16: **end for**

---

must be adopted to control the regret. As a result, there is a trade-off between regret and violation that is determined by the amount of exploration performed. In the rest of this section, we design an algorithm that is able to optimally control such a trade-off.

### 5.1 The OPPS Algorithm With Partial Feedback

We extend the OPPS algorithm introduced in Section 4 to deal with the *partial* feedback case. The idea behind the new algorithm is to split episodes into two phases. The *first* one is an exploration phase with the goal of building a sufficiently-good approximation of persuasiveness constraints, so as to achieve sublinear violation. Such a phase lasts for the first $N|X||\Omega||A|$ episodes, where we let $N := \lceil T^\alpha \rceil$ with $\alpha \in [0,1]$ being a parameter controlling the length of the two phases, given as input to the algorithm. The *second* phase is instead devoted to achieving sublinear regret, and it follows the same steps of OPPS with full feedback (Algorithm 2). The first phase works by considering each $(x, \omega, a) \in X \times \Omega \times A$ for $N$ episodes. When $(x, \omega, a)$ is considered at episode $t \in [T]$, the algorithm commits to a signaling scheme induced by an occupancy measure $\widehat{q}_t$ that maximizes the probability $\sum_{x' \in X} q(x, \omega, a, x')$ of visiting such a triplet, while at the same time satisfying all the constraints of the Opt-Opt problem. Crucially, such a procedure does *not* guarantee that every triplet

is visited $N$ times. Indeed, there might be triplets $(x, \omega, a)$ that are visited with very low probability. This can be the case when either transitions and priors place very low probability on $(x, \omega)$ or action $a$ is associated with very low receivers' rewards, and, thus, it must be recommended with very low probability to satisfy the optimistic persuasiveness constraints defined in `Opt-Opt`.

Algorithm 3 provides the pseudocode of `OPPS` with *partial* feedback. Notice that the variables $C(x, \omega, a)$ (initialized in Line 3 and updated in Line 9) are counters used to keep track of how many times each triplet $(x, \omega, a)$ is considered during the first phase, namely when $t \leq N|X||\Omega||A|$. Moreover, the algorithm ensures that every triplet is considered exactly $N$ times during the first phase, by selecting them accordingly as in Line 7. Let us also observe that Algorithm 3 updates all the estimators and bounds (by using partial feedback) and selects the signaling policy $\phi_t$ as done by Algorithm 2. The main difference with respect to Algorithm 2 is that $\widehat{q}_t$ used to define $\phi_t$ is computed in a different way during the first (exploration) phase (see Line 8).

## 5.2 Algorithm Analysis With Partial Feedback

In the following, we prove the guarantees attained by `OPPS` with *partial* feedback. We start by stating the following result on the regret attained by the algorithm.

**Theorem 4.** *Given any $\delta \in (0, 1)$, with probability at least $1 - 7\delta$, Algorithm 3 attains regret*
$$R_T \leq \widetilde{\mathcal{O}}\left( NL|X||\Omega||A| + L^2|X|\sqrt{T|A||\Omega|\ln\left(1/\delta\right)} \right).$$

In order to prove Theorem 4, we split the analysis into two cases: one targets exploration episodes in the first phase of the algorithm, while the other is concerned with the subsequent (exploitation) phase. In the first $N$ episodes in which the `OPPS` algorithm explores without being driven by the `Opt-Opt` objective, the algorithm incurs in linear regret. Instead, in the second phase, `OPPS` employs an optimistic approach, since the algorithm is driven by the objective of the `Opt-Opt` problem. Thus, in the second phase, the algorithm attains regret sublinear in $T$. The two cases combined give the regret bound provided in Theorem 4. Next, we state the result on the violations attained by `OPPS`.

**Theorem 5.** *Given any $\delta \in (0, 1)$, with probability at least $1 - 9\delta$, Algorithm 3 attains cumulative violation $V_T \leq \widetilde{\mathcal{O}}\left( \rho \left( |A|\frac{T}{\sqrt{N}} + |A|\sqrt{N} + L^2\sqrt{T} \right) \right)$, where $\rho := (|X||\Omega||A|)^{3/2}\sqrt{\ln\left(1/\delta\right)}$.*

Proving Theorem 5 requires a non-trivial analysis. The result follows by showing that uniformly exploring over feasible solutions to the `Opt-Opt` problem leads to a violation bound of the order of $\mathcal{O}(\sqrt{N})$ during the exploration phase. Intuitively, this follows by upper bounding the occupancy measure in each triplet $(x, \omega, a)$ with an occupancy of a previous (exploration) episode, relative to the best response of the follower in state $x$ upon receiving action recommendation $a$. Theorems 4 and 5 establish the trade-off between regret and violation achieved by the `OPPS` algorithm. Indeed, by recalling the definition of $N$ (see Line 1 in Algorithm 3), it is easy to see that the algorithm attains regret $R_T \leq \widetilde{\mathcal{O}}(T^\alpha)$ and violation $V_T \leq \widetilde{\mathcal{O}}(T^{1-\alpha/2})$, where $\alpha \in [1/2, 1]$ is the parameter controlling the trade-off, given as input to the algorithm.

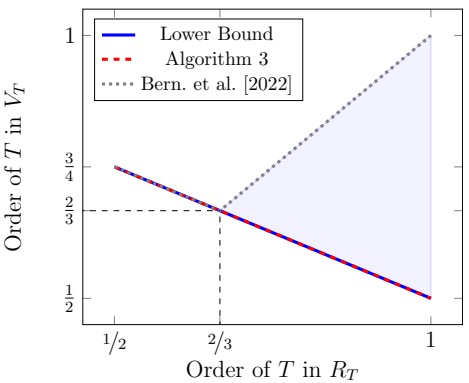

Figure 1: Trade-off between regret and violation achieved by Algorithm 3 compared with the lower bound in Theorem 6. The figure also shows the regret upper bound obtained in a related setting by Bernasconi et al. [2022], with the blue area representing the gap that we are able to close in our setting.

## 5.3 Lower Bound

We conclude the section and the paper by showing that the regret and violation bounds attained by the `OPPS` algorithm (see Theorems 4 and 5) are tight for any choice of $\alpha \in [1/2, 1]$. We do so by devising a lower bound matching these bounds (Theorem 6). Its main idea is to consider two instances of

episodic MPP involving a receiver with two actions $a_1, a_2$ such that only $a_1$ provides positive reward to the sender. In one instance, receiver's rewards by playing $a_1$ are higher than those obtained by taking $a_2$, while in the second instance the opposite holds. As a result, recommending action $a_1$ results in low regret in the first instance and high violation in the second one, while recommending action $a_2$ results in low violation in the second instance and high regret in the first one. This leads to the trade-off formally stated by the following theorem.

**Theorem 6.** *Given $\alpha \in [1/2, 1]$, there is no learning algorithm achieving both $R_T = o(T^\alpha)$ and $V_T = o(T^{1-\alpha/2})$ with probability greater or equal to a fixed constant $\psi > 0$.*

Theorem 6 shows that the bounds in Theorems 4 and 5 are tight for any $\alpha \in [1/2, 1]$. See Figure 1 for a graphical depiction. Theorem 6 also proves that it is impossible to achieve sublinear regret while being persuasive at every episode with high probability, when the sender has no information about the receivers. Notice that, in our MPP setting with partial feedback, we deal with a trade-off between regret and violation that is similar to the one faced by Bernasconi et al. [2022] in related settings. Differently from them, we are able to achieve an optimal trade-off for any $\alpha \in [1/2, 1]$. Indeed, Bernasconi et al. [2022] only obtain optimality for $\alpha \in [1/2, 2/3]$, leaving as an open problem matching the lower bound for the other values of the parameter $\alpha$ (see Figure 1). Crucially, we are able to achieve trade-off optimality by using a clever exploration method. Indeed, when considering a triplet $(x, \omega, a)$ in the first phase, the OPPS algorithm does *not* simply commit to a signaling policy that maximizes the probability of visiting such a triplet, but it rather does so while also *optimistically* accounting for persuasiveness constraints. This allows to reduce the violation cumulated during the first phase, thus achieving trade-off optimality.

## Acknowledgments

This paper is supported by the Italian MIUR PRIN 2022 Project "Targeted Learning Dynamics: Computing Efficient and Fair Equilibria through No-Regret Algorithms", by the FAIR (Future Artificial Intelligence Research) project, funded by the NextGenerationEU program within the PNRR-PE-AI scheme (M4C2, Investment 1.3, Line on Artificial Intelligence), and by the EU Horizon project ELIAS (European Lighthouse of AI for Sustainability, No. 101120237).

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

# Appendix

The appendix is organized as follows:

- In Appendix A we report the related works concerning the online learning in Markov decision processes and online Bayesian persuasion literatures.
- In Appendix B we describe the estimators and the confidence bounds related to the stochastic quantities of the Markov persuasive processes.
- In Appendix C we report the per-round optimization problem performed by the algorithms we present.
- In Appendix D we report the omitted proofs related to the *full-feedback* setting.
- In Appendix E we report the omitted proofs related to the *partial-feedback* setting.

## A    Related works

**Sequential Bayesian persuasion**    The work that is most related to ours is [Wu et al., 2022], which introduces MPPs. Specifically, Wu et al. [2022] study settings where the sender knows everything about receivers' rewards, with the only elements unknown to them being their rewards, transition probabilities, and prior distributions over outcomes. Moreover, they also assume that the receivers know everything they need about the environment, so as to select a best-response action, and that all rewards are deterministic. In contrast, we consider MPP settings in which sender and receivers have no knowledge of the environment, including their rewards, which we assume to be stochastic. Moreover, Wu et al. [2022] obtain a regret bound of the order of $\widetilde{\mathcal{O}}(\sqrt{T}/D)$, where $D$ is a parameter related to receivers' rewards. Notice that such a dependence is particularly unpleasant, as $D$ may be exponentially large in instances in which there are some receivers' actions that are best responses only for a "small" space of information-disclosure policies. Other works related to ours are [Gan et al., 2022], which studies a Bayesian persuasion problem where a sender sequentially interacts with a myopic receiver in a multi-state environment, and [Bernasconi et al., 2023b], which addresses MPPs with a farsighted receiver. These two works considerably depart from ours, as they both assume that the sender knows everything about the environment, including transitions, priors, and rewards. Thus, they are *not* concerned with learning problems, but with the problem of computing optimal information-disclosure policies. Finally, [Bernasconi et al., 2022] studies settings where a sender faces a farsighted receiver in a sequential environment with a tree structure, addressing the case in which the only elements unknown to the sender are the prior distributions over outcomes, while rewards are deterministic and known. The tree structure considerably eases the learning task, as it allows to express sender's expected rewards linearly in variables defining information-disclosure policies. Intuitively, this allows to factor the uncertainty about transitions in the rewards at the leaves of the tree.

**Online Bayesian persuasion**    It is also worth citing some works that study learning problems in which a one-shot Bayesian persuasion setting is played repeatedly [Castiglioni et al., 2020b, 2021b, Zu et al., 2021, Bernasconi et al., 2023a]. These works considerably depart from ours, since they do *not* consider any kind of sequential structure in the sender-receiver interaction at each episode.

**Online learning in constrained MDPs**    Our paper is also related to the problem of designing no-regret algorithms in online constrained Markov decision processes. The literature on online learning in Markov decision processes is extensive (see, *e.g.*, Auer et al. [2008], Even-Dar et al. [2009], Neu et al. [2010] for fundamental works on the topic). In such settings, two types of feedback are usually investigated. The *full-information feedback* setting [Rosenberg and Mansour, 2019], in which the entire reward function is observed after the learner's choice and the *partial feedback* setting [Jin et al., 2020], where the learner only observes the reward gained during the episode. Over the last decade, there has been significant attention to the field of online Markov decision processes in presence of constraints. The majority of previous works on this topic have focused on settings where constraints are stochastically sampled from a fixed distribution (see, *e.g.*, Zheng and Ratliff [2020]). Wei et al. [2018] deal with adversarial reward and stochastic constraints, assuming known transition probabilities and full information feedback. Efroni et al. [2020] propose two approaches to address the exploration-exploitation dilemma in episodic constrained MDPs. These approaches guarantee

sublinear regret and constraint violation when transition probabilities, rewards, and constraints are unknown and stochastic, and the feedback is partial. Qiu et al. [2020] provide a primal-dual approach based on *optimism in the face of uncertainty*. This work shows the effectiveness of such an approach when dealing with episodic constrained MDPs with adversarial rewards and stochastic constraints, achieving both sublinear regret and constraint violation with full-information feedback. Finally, Stradi et al. [2024] propose a best-of-both-worlds algorithm in constrained Markov decision processes with full information feedback. While the previous works are related to ours, the aforementioned techniques cannot be easily generalized to our setting as they are not designed to properly handle the presence of outcomes and IC constraints.

## B  Confidence bounds

In this section, we further describe the estimators and confidence bounds for the stochastic quantities involved in an episodic MPP, namely, transitions, priors, sender's rewards, and receivers' ones.

### B.1  Transition probabilities

First, we introduce confidence bounds for transition probabilities $P(x'|x, \omega, a)$, by generalizing those introduced by Rosenberg and Mansour [2019] for MDPs to MPPs. In the following, we let $N_t(x, \omega, a)$, respectively $N_t(x, \omega, a, x')$, be the counter specifying the number of episodes up to episode $t \in [T]$ (excluded) in which the triplet $(x, \omega, a)$, respectively the tuple $(x, \omega, a, x')$, is visited. Then, the estimated probability of going from $x \in X$ to $x' \in X$ by playing action $a \in A$, when the outcome realized in state $x$ is $\omega \in \Omega$, is defined as follows:

$$\overline{P}_t\left(x'|x, \omega, a\right) := \frac{N_t(x, \omega, a, x')}{\max\left\{1, N_t(x, \omega, a)\right\}}.$$

For any $\delta \in (0, 1)$, the confidence set at episode $t \in [T]$ for the transition function $P$ is $\mathcal{P}_t := \bigcap_{(x, \omega, a) \in X \times \Omega \times A} \mathcal{P}_t^{x, \omega, a}$, where $\mathcal{P}_t^{x, \omega, a}$ is a set of transition functions defined as:

$$\mathcal{P}_t^{x, \omega, a} := \left\{\widehat{P} : \left\|\widehat{P}(\cdot|x, \omega, a) - \overline{P}_t(\cdot|x, \omega, a)\right\|_1 \leq \epsilon_t(x, \omega, a)\right\},$$

where $\widehat{P}(\cdot|x, \omega, a)$ and $\overline{P}_t(\cdot|x, \omega, a)$ are vectors whose entries are the values $\widehat{P}(x'|x, \omega, a)$ and $\overline{P}_t(x'|x, \omega, a)$, respectively, while $\epsilon_t(x, \omega, a)$ is a confidence bound defined as:

$$\epsilon_t(x, \omega, a) := \sqrt{\frac{2|X_{k(x)+1}| \ln\left(T|X||\Omega||A|/\delta\right)}{\max\left\{1, N_t(x, \omega, a)\right\}}}.$$

The following lemma formally proves that $\mathcal{P}_t$ is a suitable confidence set for the transition function of an MPP.

**Lemma 4.** *Given any $\delta \in (0, 1)$, with probability at least $1 - \delta$, the following condition holds for every $x \in X$, $\omega \in \Omega$, $a \in A$, and $t \in [T]$ jointly:*

$$\left\|P(\cdot|x, \omega, a) - \overline{P}_t(\cdot|x, \omega, a)\right\|_1 \leq \epsilon_t(x, \omega, a).$$

Lemma 4 can be easily proven by applying the same analysis as presented in [Auer et al., 2008] and employing a union bound over all $x$, $\omega$, $a$, and $t$..

### B.2  Prior distributions

Next, we introduce confidence bounds for prior distributions. For every state $x \in X$, we define $\overline{\mu}_t(\cdot|x) \in \Delta(\Omega)$ as the estimator of the prior distribution at $x$ built by using observations up to episode $t \in [T]$ (this excluded). Formally, the entries of vector $\overline{\mu}_t(\cdot|x)$ are such that, for every $\omega \in \Omega$:

$$\overline{\mu}_t(\omega|x) := \frac{\sum_{\tau \in [t-1]} \mathbb{1}_\tau\{x, \omega\}}{\max\{1, N_t(x)\}},$$

where $N_t(x)$ is the number of visits to state $x$ up to episode $t$ (excluded), while $\mathbb{1}_\tau\{x, \omega\}$ is an indicator function equal to 1 if and only if the pair $(x, \omega)$ is visited at episode $\tau$.

The following lemma provides confidence bounds for priors.

**Lemma 5.** *Given any $\delta \in (0, 1)$, with probability at least $1 - \delta$, the following holds for all $x \in X$ and $t \in [T]$ jointly:*

$$\|\mu(\cdot|x) - \overline{\mu}_t(\cdot|x)\|_1 \leq \zeta_t(x),$$

*where we let $\zeta_t(x) := \sqrt{\frac{2|\Omega| \ln(T|X|/\delta)}{\max\{1, N_t(x)\}}}$.*

Lemma 5 follows by applying Bernstein's inequality and a union bound over all states and episodes.

### B.3 Sender's and receivers' rewards

Finally, we introduce estimators for rewards. In the following, present the results related to sender's rewards and receiver's rewards under full and partial feedback. For every $x \in X$, $\omega \in \Omega$, and $a \in A$, the estimated sender's and receivers' rewards built with observations up to episode $t \in [T]$ (this excluded) are defined as follows:

$$\overline{r}_{S,t}(x, \omega, a) := \frac{\sum_{\tau \in [t-1]} r_{S,\tau}(x, \omega, a) \mathbb{1}_\tau \{x, \omega, a\}}{\max\{1, N_t(x, \omega, a)\}},$$

$$\overline{r}_{R,t}(x, \omega, a) := \frac{\sum_{\tau \in [t-1]} r_{R,\tau}(x, \omega, a) \mathbb{1}_\tau \{x, \omega, a\}}{\max\{1, N_t(x, \omega, a)\}},$$

where $\mathbb{1}_\tau \{x, \omega, a\}$ is an indicator function equal to 1 if and only if the triplet $(x, \omega, a)$ is visited during episode $\tau$.

The following lemma provides confidence bounds for sender's rewards, when *full* feedback is available.

**Lemma 6.** *Given any $\delta \in (0, 1)$, with probability at least $1 - \delta$, the following condition holds for every $x \in X$, $\omega \in \Omega$, $a \in A$, and $t \in [T]$ jointly:*

$$\left| r_S(x, \omega, a) - \overline{r}_{S,t}(x, \omega, a) \right| \leq \xi_{S,t}(x, \omega, a),$$

*where $\xi_{S,t}(x, \omega, a) := \min\left\{1, \sqrt{\frac{\ln(3T|X||\Omega|/\delta)}{\max\{1, N_t(x, \omega)\}}}\right\}$.*

Lemma 6 follows by applying Hoeffding's inequality and a union bound over all $x$, $\omega$ and $t$.

The following lemma provides confidence bounds for receiver's rewards, when *full* feedback is available.

**Lemma 7.** *Given any $\delta \in (0, 1)$, with probability at least $1 - \delta$, the following condition holds for every $x \in X$, $\omega \in \Omega$, $a \in A$, and $t \in [T]$ jointly:*

$$\left| r_R(x, \omega, a) - \overline{r}_{R,t}(x, \omega, a) \right| \leq \xi_{R,t}(x, \omega, a),$$

*where $\xi_{R,t}(x, \omega, a) := \min\left\{1, \sqrt{\frac{\ln(3T|X||\Omega|/\delta)}{\max\{1, N_t(x, \omega)\}}}\right\}$.*

Lemma 7 follows by applying Hoeffding's inequality and a union bound over all $x$, $\omega$ and $t$.

The following lemma provides confidence bounds for sender's rewards, when only *partial* feedback is available.

**Lemma 8.** *Given any $\delta \in (0, 1)$, with probability at least $1 - \delta$, the following condition holds for every $x \in X$, $\omega \in \Omega$, $a \in A$, and $t \in [T]$ jointly:*

$$\left| r_S(x, \omega, a) - \overline{r}_{S,t}(x, \omega, a) \right| \leq \xi_{S,t}(x, \omega, a),$$

*where $\xi_{S,t}(x, \omega, a) := \min\left\{1, \sqrt{\frac{\ln(3T|X||\Omega||A|/\delta)}{\max\{1, N_t(x, \omega, a)\}}}\right\}$.*

Lemma 8 follows by applying Hoeffding's inequality and a union bound over all $x$, $\omega$, $a$, and $t$.

Finally, the following lemma provides confidence bounds for receiver's rewards, when only *partial* feedback is available.

**Lemma 9.** *Given any $\delta \in (0,1)$, with probability at least $1 - \delta$, the following condition holds for every $x \in X$, $\omega \in \Omega$, $a \in A$, and $t \in [T]$ jointly:*

$$\left| r_R(x,\omega,a) - \overline{r}_{R,t}(x,\omega,a) \right| \leq \xi_{R,t}(x,\omega,a),$$

*where* $\xi_{R,t}(x,\omega,a) := \min \left\{ 1, \sqrt{\frac{\ln(3T|X||\Omega||A|/\delta)}{\max\{1,N_t(x,\omega,a)\}}} \right\}.$

Lemma 9 follows by applying Hoeffding's inequality and a union bound over all $x$, $\omega$, $a$, and $t$.

## C   Optimistic optimization problem

In the following section we describe the linear program solved by Algorithm 2 and Algorithm 3, namely `Opt-Opt`. Intuitively, `Opt-Opt` is the optimistic version of Program (1), since the objective is guided by the optimism and the confidence bounds of the estimated parameters are chosen to make constraints easier to be satisfied. Notice that the confidence bounds on the transitions and the prior are applied to the $\| \cdot \|_1$ differences between the empirical and the real mean of the distributions. Thus, in order to insert the aforementioned confidence bounds in a LP-formulation, the related constraints must be linearized by means of additional optimization variables.

The linear program solved by Algorithm 2 and Algorithm 3 is the following.

$$\max_{q,\zeta,\epsilon} \quad \sum_{x \in X_k} \sum_{\omega \in \Omega} \sum_{a \in A} \sum_{x' \in X_{k+1}} q(x,\omega,a,x')\Big( \overline{r}_{S,t}(x,\omega,a) + \xi_{S,t}(x,\omega,a) \Big) \quad \text{s.t.} \tag{2a}$$

$$\sum_{x \in X_k} \sum_{\omega \in \Omega} \sum_{a \in A} \sum_{x' \in X_{k+1}} q(x,\omega,a,x') = 1 \qquad\qquad \forall k \in [0 \dots L-1] \tag{2b}$$

$$\sum_{x' \in X_{k-1}} \sum_{\omega \in \Omega} \sum_{a \in A} q(x',\omega,a,x) = \sum_{\omega \in \Omega} \sum_{a \in A} \sum_{x' \in X_{k+1}} q(x,\omega,a,x')$$

$$\forall k \in [0 \dots L-1], \forall x \in X_k \tag{2c}$$

$$q(x,\omega,a,x') - \overline{P}_t(x'|x,\omega,a) \sum_{x'' \in X_{k+1}} q(x,\omega,a,x'') \leq \epsilon(x,\omega,a,x')$$

$$\forall k \in [0 \dots L-1], \forall (x,\omega,a,x') \in X_k \times \Omega \times A \times X_{k+1} \tag{2d}$$

$$\overline{P}_t(x'|x,a,\omega) \sum_{x'' \in X_{k+1}} q(x,\omega,a,x'') - q(x,\omega,a,x') \leq \epsilon(x,\omega,a,x')$$

$$\forall k \in [0 \dots L-1], \forall (x,\omega,a,x') \in X_k \times \Omega \times A \times X_{k+1} \tag{2e}$$

$$\sum_{x' \in X_{k+1}} \epsilon(x,\omega,a,x') \leq \epsilon_t(x,\omega,a) \sum_{x' \in X_{k+1}} q(x,\omega,a,x')$$

$$\forall k \in [0 \dots L-1], \forall (x,\omega,a) \in X_k \times \Omega \times A \tag{2f}$$

$$q(x,\omega) - \overline{\mu}_t(\omega|x) \sum_{\omega' \in \Omega} q(x,\omega') \leq \zeta(x,\omega) \qquad \forall k \in [0 \dots L-1], \forall (x,\omega) \in X_k \times \Omega \tag{2g}$$

$$\overline{\mu}_t(\omega|x) \sum_{\omega' \in \Omega} q(x,\omega') - q(x,\omega) \leq \zeta(x,\omega) \qquad \forall k \in [0 \dots L-1], \forall (x,\omega) \in X_k \times \Omega \tag{2h}$$

$$\sum_{\omega \in \Omega} \zeta(x,\omega) \leq \zeta_t(x) \sum_{\omega \in \Omega} q(x,\omega), \qquad\qquad \forall k \in [0 \dots L-1], \forall x \in X_k \tag{2i}$$

$$\sum_{\omega \in \Omega} \sum_{x' \in X_{k+1}} q(x,\omega,a,x')\Big( \overline{r}_{R,t}(x,\omega,a) + \xi_{R,t}(x,\omega,a)$$

$$- \overline{r}_{R,t}(x,\omega,a') + \xi_{R,t}(x,\omega,a') \Big) \geq 0$$

$$\forall k \in [0 \dots L-1], \forall (x,a) \in X_k \times A, \forall a' \in A \tag{2j}$$

$$q(x,\omega,a,x') \geq 0 \qquad\qquad \forall k \in [0 \dots L-1], \forall (x,a,x') \in X_k \times \Omega \times A \times X_{k+1}, \tag{2k}$$

where Objective (2a) maximizes the upper confidence bound of the sender reward, Constraint (2b) ensures that the occupancy measure sums to 1 for every layer, Constraint (2c) is the flow constraint, Constraint (2d) is related to the confidence interval on the transition functions, Constraint (2e) is still related to the confidence bounds on the transition function, Constraint (2f) allows to write linearly the constraints related to the transition functions even if the interval holds for the $\|\cdot\|_1$, Constraint (2g) is related to the confidence interval on the outcomes, Constraint (2h) is still related to the confidence bounds on the outcomes, Constraint (2i) allows to write linearly the constraints related to the outcomes even if the interval holds for the $\|\cdot\|_1$, Constraint (2j) is the optimistic constraint for the Incentive Compatibility (IC) property and, finally, Constraint (2k) ensures that the occupancy are greater than zero.

**Lemma 2.** *Given $\delta \in (0,1)$, under event $\mathcal{E}(\delta)$, Opt-Opt admits a feasible solution at every $t \in [T]$.*

*Proof.* First we notice that under the clean event $\mathcal{E}(\delta)$ the true transition function $P$ and the prior $\mu$ are included in the their confidence interval; thus, they are available in the constrained space defined by Opt-Opt. Then, we focus on the incentive compatibility constraints. Referring as $q^\diamond$ to an incentive compatible occupancy measure, under $\mathcal{E}(\delta)$, we have that:

$$\sum_{\omega \in \Omega, x' \in X_{k+1}} q^\diamond(x, \omega, a, x') \left( \overline{r}_{R,t}(x, \omega, a) + \xi_{R,t}(x, \omega, a) - \overline{r}_{R,t}(x, \omega, \overline{a}) + \xi_{R,t}(x, \omega, \overline{a}) \right) \geq$$

$$\sum_{\omega \in \Omega, x' \in X_{k+1}} q^\diamond(x, \omega, a, x') \left( r_R(x, \omega, a) - r_R(x, \omega, \overline{a}) \right) \geq 0,$$

for any $k \in [L-1], (x,a) \in X_k \times A, \forall \overline{a} \in A$. As a result, if $q^\diamond$ is incentive compatible, it belongs to the optimistic decision space, which concludes the proof. $\square$

# D   Full feedback

In this section we report the omitted proof related to Algorithm 2. Notice that the bound on the transition function estimations still hold when the feedback is partial.

## D.1   Transition functions

We start by showing that the estimated occupancy measures which encompass the information related to the outcomes and the transitions concentrate with respect to the true occupancy measures.

**Lemma 3.** *Given any $\delta \in (0,1)$, under the clean event $\mathcal{E}(\delta)$, with probability at least $1 - 2\delta$, it holds that $\sum_{t \in [T]} \|q_t - \widehat{q}_t\|_1 \leq \mathcal{O}\left( L^2 |X| \sqrt{T|A||\Omega| \ln\left( T|X||\Omega||A|/\delta \right)} \right).$*

*Proof.* We start noticing that, for any $(x, \omega, a) \in X \times \Omega \times A$, we have:

$$\sum_{x' \in X_{k(x)+1}} \left| q^{P_t, \phi_t, \mu_t}(x, \omega, a, x') - q^{P, \phi_t, \mu}(x, \omega, a, x') \right|$$

$$= \sum_{x' \in X_{k(x)+1}} \left| q^{P_t, \phi_t, \mu_t}(x, \omega, a) P_t(x'|x, \omega, a) - q^{P, \phi_t, \mu}(x, \omega, a) P(x'|x, \omega, a) \right|$$

$$\leq \sum_{x' \in X_{k(x)+1}} \left| q^{P_t, \phi_t, \mu_t}(x, \omega, a) P_t(x'|x, \omega, a) - q^{P, \phi_t, \mu}(x, \omega, a) P_t(x'|x, \omega, a) \right|$$

$$+ \sum_{x' \in X_{k(x)+1}} \left| q^{P, \phi_t, \mu}(x, \omega, a) P_t(x'|x, \omega, a) - q^{P, \phi_t, \mu}(x, \omega, a) P(x'|x, \omega, a) \right|$$

$$= \sum_{x' \in X_{k(x)+1}} \left| q^{P_t, \phi_t, \mu_t}(x, \omega, a) - q^{P, \phi_t, \mu}(x, \omega, a) \right| P_t(x'|x, \omega, a)$$

$$+ \sum_{x' \in X_{k(x)+1}} q^{P, \phi_t, \mu}(x, \omega, a) \left| P_t(x'|x, \omega, a) - P(x'|x, \omega, a) \right|$$

$$= \left| q^{P_t, \phi_t, \mu_t}(x, \omega, a) - q^{P, \phi_t, \mu}(x, \omega, a) \right| + q^{P, \phi_t, \mu}(x, \omega, a) \| P_t(\cdot|x, \omega, a) - P(\cdot|x, \omega, a) \|_1.$$

Thus, summing over $t \in [T]$ and $(x, \omega, a) \in X \times \Omega \times A$ we obtain:

$$\sum_{t \in [T]} \|q_t - \widehat{q}_t\|_1 \leq \sum_{t \in [T]} \sum_{x \in X, \omega \in \Omega, a \in A} \Big( \big| q^{P_t, \phi_t, \mu_t}(x, \omega, a) - q^{P, \phi_t, \mu}(x, \omega, a) \big|$$

$$+ q^{P, \phi_t, \mu}(x, \omega, a) \| P_t(\cdot | x, \omega, a) - P(\cdot | x, \omega, a) \|_1 \Big).$$

Next, we focus on the first part of the equation, noticing that:

$$|q^{P_t, \phi_t, \mu_t}(x, \omega, a) - q^{P, \phi_t, \mu}(x, \omega, a)|$$
$$\leq \big| q^{P_t, \phi_t, \mu_t}(x, \omega, a) - q^{P_t, \phi_t, \mu}(x, \omega, a) \big| + \big| q^{P_t, \phi_t, \mu}(x, \omega, a) - q^{P, \phi_t, \mu}(x, \omega, a) \big|$$

**Bound on** $\big| q^{P_t, \phi_t, \mu_t}(x, \omega, a) - q^{P_t, \phi_t, \mu}(x, \omega, a) \big|$    We bound this term by induction. At the first layer we have:

$$\sum_{x_0 \in X_0} \sum_{\omega_0 \in \Omega} \sum_{a_0 \in A} \big| q^{P_t, \phi_t, \mu_t}(x_0, \omega_0, a_0) - q^{P_t, \phi_t, \mu}(x_0, \omega_0, a_0) \big|$$

$$= \sum_{\omega_0 \in \Omega} \sum_{a_0 \in A} |\mu_t(x_0, \omega_0) \phi_t(a_0 | x_0, \omega_0) - \mu(x_0, \omega_0) \phi_t(a_0 | x_0, \omega_0)|$$

$$\leq \sum_{\omega_0 \in \Omega} |\mu_t(x_0, \omega_0) - \mu(x_0, \omega_0)|$$

$$= q^{P_t, \phi_t, \mu}(x_0) \sum_{\omega_0 \in \Omega} |\mu_t(x_0, \omega_0) - \mu(x_0, \omega_0)|.$$

observing that $X_0 = \{x_0\}$. Now we show that, if the result holds for $x_{k-1}$, it holds for $x_k$, as follows,

$$\sum_{x_k \in X_k} \sum_{\omega_k \in \Omega} \sum_{a_k \in A} \big| q^{P_t, \phi_t, \mu_t}(x_k, \omega_k, a_k) - q^{P_t, \phi_t, \mu}(x_k, \omega_k, a_k) \big|$$

$$= \sum_{x_{k-1} \in X_{k-1}} \sum_{\omega_{k-1} \in \Omega} \sum_{a_{k-1} \in A} \sum_{x_k \in X_k} \sum_{\omega_k \in \Omega} \sum_{a_k \in A} |q^{P_t, \phi_t, \mu_t}(x_{k-1}, \omega_{k-1}, a_{k-1}) \cdot$$

$$\cdot P_t(x_k | x_{k-1}, \omega_{k-1}, a_{k-1}) \mu_t(x_k, \omega_k) +$$

$$- q^{P_t, \phi_t, \mu}(x_{k-1}, \omega_{k-1}, a_{k-1}) P_t(x_k | x_{k-1}, \omega_{k-1}, a_{k-1}) \mu(x_k, \omega_k)| \phi_t(a_k | x_k, \omega_k)$$

$$= \sum_{x_{k-1} \in X_{k-1}} \sum_{\omega_{k-1} \in \Omega} \sum_{a_{k-1} \in A} \sum_{x_k \in X_k} \sum_{\omega_k \in \Omega} |q^{P_t, \phi_t, \mu_t}(x_{k-1}, \omega_{k-1}, a_{k-1}) \cdot$$

$$\cdot P_t(x_k | x_{k-1}, \omega_{k-1}, a_{k-1}) \mu_t(x_k, \omega_k) +$$

$$- q^{P_t, \phi_t, \mu}(x_{k-1}, \omega_{k-1}, a_{k-1}) P_t(x_k | x_{k-1}, \omega_{k-1}, a_{k-1}) \mu(x_k, \omega_k)|$$

$$\leq \sum_{x_{k-1} \in X_{k-1}} \sum_{\omega_{k-1} \in \Omega} \sum_{a_{k-1} \in A} \sum_{x_k \in X_k} \sum_{\omega_k \in \Omega} |q^{P_t, \phi_t, \mu_t}(x_{k-1}, \omega_{k-1}, a_{k-1}) \cdot$$

$$\cdot P_t(x_k | x_{k-1}, \omega_{k-1}, a_{k-1}) \mu_t(x_k, \omega_k) +$$

$$- q^{P_t, \phi_t, \mu}(x_{k-1}, \omega_{k-1}, a_{k-1}) P_t(x_k | x_{k-1}, \omega_{k-1}, a_{k-1}) \mu_t(x_k, \omega_k)| +$$

$$+ \sum_{x_{k-1} \in X_{k-1}} \sum_{\omega_{k-1} \in \Omega} \sum_{a_{k-1} \in A} \sum_{x_k \in X_k} \sum_{\omega_k \in \Omega} |q^{P_t, \phi_t, \mu}(x_{k-1}, \omega_{k-1}, a_{k-1}) \cdot$$

$$\cdot P_t(x_k | x_{k-1}, \omega_{k-1}, a_{k-1}) \mu_t(x_k, \omega_k) +$$

$$- q^{P_t, \phi_t, \mu}(x_{k-1}, \omega_{k-1}, a_{k-1}) P_t(x_k | x_{k-1}, \omega_{k-1}, a_{k-1}) \mu(x_k, \omega_k)|$$

$$\leq \sum_{x_{k-1} \in X_{k-1}} \sum_{k \ \omega_{k-1} \in \Omega} \sum_{a_{k-1} \in A} \big| q^{P_t, \phi_t, \mu_t}(x_{k-1}, \omega_{k-1}, a_{k-1}) - q^{P_t, \phi_t, \mu}(x_{k-1}, \omega_{k-1}, a_{k-1}) \big|$$

$$+ \sum_{x_k \in X_k} q^{P_t, \phi_t, \mu}(x_k) \sum_{\omega_k \in \Omega} |\mu_t(x_k, \omega_k) - \mu(x_k, \omega_k)|.$$

Thus, by induction hypothesis, it follows,

$$\sum_{x_k \in X_k} \sum_{\omega_k \in \Omega} \sum_{a_k \in A} \big| q^{P_t, \phi_t, \mu_t}(x_k, \omega_k, a_k) - q^{P_t, \phi_t, \mu}(x_k, \omega_k, a_k) \big|$$

$$\leq \sum_{s=0}^{k} \sum_{x_s \in X_s} q^{P_t, \phi_t, \mu}(x_s) \|\mu_t(\cdot | x_s) - \mu(\cdot | x_s)\|_1.$$

**Bound on** $\left| q^{P_t, \phi_t, \mu}(x, \omega, a) - q^{P, \phi_t, \mu}(x, \omega, a) \right|$ To bound this term, we proceed again by induction. Thus, we notice that:

$$\sum_{x_1 \in X_1} \sum_{\omega_1 \in \Omega} \sum_{a_1 \in A} |q^{P_t, \phi_t, \mu}(x_1, \omega_1, a_1) - q^{P, \phi_t, \mu}(x_1, \omega_1, a_1)|$$

$$= \sum_{\omega_0 \in \Omega} \sum_{a_0 \in A} \sum_{x_1 \in X_1} \sum_{\omega_1 \in \Omega} \sum_{a_1 \in A} |\mu(x_0, \omega_0) \phi_t(a_0 | x_0, \omega_0) P_t(x_1 | x_0, \omega_0, a_0) \mu(x_1, \omega_1) \phi_t(a_1 | x_1, \omega_1)$$

$$- \mu(x_0, \omega_0) \phi_t(a_0 | x_0, \omega_0) P(x_1 | x_0, \omega_0, a_0) \mu(x_1, \omega_1) \phi_t(a_1 | x_1, \omega_1)|$$

$$= \sum_{\omega_0 \in \Omega} \sum_{a_0 \in A} \mu(x_0, \omega_0) \phi_t(a_0 | x_0, \omega_0) \sum_{x_1 \in X_1} |P_t(x_1 | x_0, \omega_0, a_0) - P(x_1 | x_0, \omega_0, a_0)| \cdot$$

$$\cdot \sum_{\omega_1 \in \Omega} \sum_{a_1 \in A} \mu(x_1, \omega_1) \phi_t(a_1 | x_1, \omega_1)|$$

$$\leq \sum_{\omega_0 \in \Omega} \sum_{a_0 \in A} q^{P, \phi_t, \mu}(x_0, \omega_0, a_0) \|P_t(\cdot | x_0, \omega_0, a_0) - P(\cdot | x_0, \omega_0, a_0)\|_1.$$

Now, we proceed with the induction step,

$$\sum_{x_k \in X_k} \sum_{\omega_k \in \Omega} \sum_{a_k \in A} |q^{P_t, \phi_t, \mu}(x_k, \omega_k, a_k) - q^{P, \phi_t, \mu}(x_k, \omega_k, a_k)|$$

$$= \sum_{x_{k-1} \in X_{k-1}} \sum_{\omega_{k-1} \in \Omega} \sum_{a_{k-1} \in A} \sum_{x_k \in X_k} \sum_{\omega_k \in \Omega} \sum_{a_k \in A} |q^{P_t, \phi_t, \mu}(x_{k-1}, \omega_{k-1}, a_{k-1}) \cdot$$

$$\cdot P_t(x_k | x_{k-1}, \omega_{k-1}, a_{k-1}) \mu(x_k, \omega_k) \phi_t(a_k | x_k, \omega_k) +$$

$$- q^{P, \phi_t, \mu}(x_{k-1}, \omega_{k-1}, a_{k-1}) P(x_k | x_{k-1}, \omega_{k-1}, a_{k-1}) \mu(x_k, \omega_k) \phi_t(a_k | x_k, \omega_k)|$$

$$= \sum_{x_{k-1} \in X_{k-1}} \sum_{\omega_{k-1} \in \Omega} \sum_{a_{k-1} \in A} \sum_{x_k \in X_k} |q^{P_t, \phi_t, \mu}(x_{k-1}, \omega_{k-1}, a_{k-1}) P_t(x_k | x_{k-1}, \omega_{k-1}, a_{k-1})$$

$$- q^{P, \phi_t, \mu}(x_{k-1}, \omega_{k-1}, a_{k-1}) P(x_k | x_{k-1}, \omega_{k-1}, a_{k-1})|$$

$$\leq \sum_{x_{k-1} \in X_{k-1}} \sum_{\omega_{k-1} \in \Omega} \sum_{a_{k-1} \in A} \sum_{x_k \in X_k} |q^{P_t, \phi_t, \mu}(x_{k-1}, \omega_{k-1}, a_{k-1}) P_t(x_k | x_{k-1}, \omega_{k-1}, a_{k-1})$$

$$- q^{P, \phi_t, \mu}(x_{k-1}, \omega_{k-1}, a_{k-1}) P_t(x_k | x_{k-1}, \omega_{k-1}, a_{k-1})|$$

$$+ \sum_{x_{k-1} \in X_{k-1}} \sum_{\omega_{k-1} \in \Omega} \sum_{a_{k-1} \in A} \sum_{x_k \in X_k} |q^{P, \phi_t, \mu}(x_{k-1}, \omega_{k-1}, a_{k-1}) P_t(x_k | x_{k-1}, \omega_{k-1}, a_{k-1})$$

$$- q^{P, \phi_t, \mu}(x_{k-1}, \omega_{k-1}, a_{k-1}) P(x_k | x_{k-1}, \omega_{k-1}, a_{k-1})|$$

$$\leq \sum_{x_{k-1} \in X_{k-1}} \sum_{\omega_{k-1} \in \Omega} \sum_{a_{k-1} \in A} |q^{P_t, \phi_t, \mu}(x_{k-1}, \omega_{k-1}, a_{k-1}) - q^{P, \phi_t, \mu}(x_{k-1}, \omega_{k-1}, a_{k-1})|$$

$$+ \sum_{x_{k-1} \in X_{k-1}} \sum_{\omega_{k-1} \in \Omega} \sum_{a_{k-1} \in A} q^{P, \phi_t, \mu}(x_{k-1}, \omega_{k-1}, a_{k-1}) \cdot$$

$$\cdot \|P_t(\cdot | x_{k-1}, \omega_{k-1}, a_{k-1}) - P(\cdot | x_{k-1}, \omega_{k-1}, a_{k-1})\|_1.$$

Thus by induction hypothesis we obtain,

$$\sum_{x_k \in X_k} \sum_{\omega_k \in \Omega} \sum_{a_k \in A} |q^{P_t, \phi_t, \mu}(x_k, \omega_k, a_k) - q^{P, \phi_t, \mu}(x_k, \omega_k, a_k)|$$

$$\leq \sum_{s=0}^{k-1} \sum_{x_s \in X_s} \sum_{\omega_s \in \Omega} \sum_{a_s \in A} q^{P, \phi_t, \mu}(x_s, \omega_s, a_s) \|P_t(\cdot | x_s, \omega_s, a_s) - P(\cdot | x_s, \omega_s, a_s)\|_1.$$

Returning to the quantity of interest we have:

$$\sum_{t \in [T]} \|q_t - \widehat{q}_t\|_1 \leq 2 \sum_{t \in [T]} \sum_{k=0}^{L-1} \sum_{s=0}^{k-1} \sum_{x_s \in X_s} \sum_{\omega_s \in \Omega} \sum_{a_s \in A} q^{P, \phi_t, \mu}(x_s, \omega_s, a_s) \|P_t(\cdot | x_s, \omega_s, a_s) +$$

$$-P(\cdot|x_s,\omega_s,a_s)\|_1 + \sum_{t\in[T]}\sum_{k=0}^{L-1}\sum_{s=0}^{k}\sum_{x_s\in X_s}q^{P_t,\phi_t,\mu}(x_s)\|\mu_t(\cdot|x_s)-\mu(\cdot|x_s)\|_1. \quad (3)$$

We proceed bounding the first term in Inequality (3). Fixing a layer $k\in[0,\dots,L-1]$, employing Azuma-Hoeffding inequality and noticing that $\|P_t(\cdot|x_k,\omega_k,a_k)-P(\cdot|x_k,\omega_k,a_k)\|_1\le 2$, we have, with probability $1-2\delta$:

$$\sum_{t\in[T]}\sum_{s=0}^{k-1}\sum_{x_s\in X_s}\sum_{\omega_s\in\Omega}\sum_{a_s\in A}q^{P,\phi_t,\mu}(x_s,\omega_s,a_s)\|P_t(\cdot|x_s,\omega_s,a_s)-P(\cdot|x_s,\omega_s,a_s)\|_1$$

$$\le\sum_{s=0}^{k-1}\sum_{t\in[T]}\sum_{x_s\in X_s}\sum_{\omega_s\in\Omega}\sum_{a_s\in A}\sqrt{\frac{2|X_{k(x_s)+1}|\ln\left(\frac{T|X||\Omega||A|}{\delta}\right)}{\max\{1,N_t(x_s,\omega_s,a_s)\}}}\mathbb{1}_t\{x_s,a_s,\omega_s\}+\sum_{s=0}^{k-1}2|X_s|\sqrt{2T\ln\left(\frac{L}{\delta}\right)}$$

$$\le\sum_{s=0}^{k-1}\sqrt{2T|X_s||X_{s+1}||A||\Omega|\ln\left(\frac{T|X||\Omega||A|}{\delta}\right)}+\sum_{s=0}^{k-1}2|X_s|\sqrt{2T\ln\left(\frac{L}{\delta}\right)}$$

$$\le|X|\sqrt{2T|A||\Omega|\ln\left(\frac{T|X||\Omega||A|}{\delta}\right)}+2|X|\sqrt{2T\ln\left(\frac{L}{\delta}\right)}.$$

Finally summing over $L$, we have, with probability at least $1-2\delta$ (which derives from a union bound between Azuma-Hoeffding inequality and the bound on the transitions):

$$\sum_{t\in[T]}\sum_{k=0}^{L-1}\sum_{s=0}^{k-1}\sum_{x_s\in X_s}\sum_{\omega_s\in\Omega}\sum_{a_s\in A}q^{P,\phi_t,\mu}(x_s,\omega_s,a_s)\|P_t(\cdot|x_s,\omega_s,a_s)-P(\cdot|x_s,\omega_s,a_s)\|_1$$

$$\le L|X|\sqrt{2T|A||\Omega|\ln\left(\frac{T|X||\Omega||A|}{\delta}\right)}+2L|X|\sqrt{2T\ln\left(\frac{L}{\delta}\right)}.$$

To bound the remaining term in Inequality (3), we proceed as follows,

$$\sum_{t\in[T]}\sum_{k=0}^{L-1}\sum_{s=0}^{k}\sum_{x_s\in X_s}q^{P_t,\phi_t,\mu}(x_s)\|\mu_t(\cdot|x_s)-\mu(\cdot|x_s)\|_1$$

$$\le\sum_{t\in[T]}\sum_{k=0}^{L-1}\sum_{s=0}^{k}\sum_{x_s\in X_s}q^{P,\phi_t,\mu}(x_s)\|\mu_t(\cdot|x_s)-\mu(\cdot|x_s)\|_1+$$

$$+\sum_{t\in[T]}\sum_{k=0}^{L-1}\sum_{s=0}^{k}\sum_{x_s\in X_s}\left(q^{P_t,\phi_t,\mu}(x_s)-q^{P,\phi_t,\mu}(x_s)\right)\|\mu_t(\cdot|x_s)-\mu(\cdot|x_s)\|_1$$

$$\le\sum_{t\in[T]}\sum_{k=0}^{L-1}\sum_{s=0}^{k}\sum_{x_s\in X_s}q^{P,\phi_t,\mu}(x_s)\|\mu_t(\cdot|x_s)-\mu(\cdot|x_s)\|_1+$$

$$+\sum_{t\in[T]}\sum_{k=0}^{L-1}\sum_{s=0}^{k}\sum_{x_s\in X_s}2\left(q^{P_t,\phi_t,\mu}(x_s)-q^{P,\phi_t,\mu}(x_s)\right)$$

$$\le\sum_{t\in[T]}\sum_{k=0}^{L-1}\sum_{s=0}^{k}\sum_{x_s\in X_s}q^{P,\phi_t,\mu}(x_s)\|\mu_t(\cdot|x_s)-\mu(\cdot|x_s)\|_1+$$

$$+\sum_{t\in[T]}\sum_{k=0}^{L-1}\sum_{s=0}^{k}\sum_{x_s\in X_s}\sum_{\omega_s\in\Omega}\sum_{a_s\in A}2\left|q^{P_t,\phi_t,\mu}(x_s,\omega_s,a_s)-q^{P,\phi_t,\mu}(x_s,\omega_s,a_s)\right|.$$

The second term is bounded by the previous analysis paying an additional $L$ factor, while, to bound the first terms we apply the Azuma-Hoeffding inequality and proceed as follows:

$$\sum_{t\in[T]}\sum_{k=0}^{L-1}\sum_{s=0}^{k}\sum_{x_s\in X_s}q^{P,\phi_t,\mu}(x_s)\sum_{\omega_s\in\Omega}|\mu_t(x_s,\omega_s)-\mu(x_s,\omega_s)|$$

$$\leq \sum_{t\in[T]}\sum_{k=0}^{L-1}\sum_{s=0}^{k}\sum_{x_s\in X_s} \mathbb{1}_t\{x_s\}\|\mu_t(\cdot|x_s)-\mu(\cdot|x_s)\|_1 + 2L|X|\sqrt{2T\ln\left(\frac{L}{\delta}\right)}$$

$$\leq \sum_{t\in[T]}\sum_{k=0}^{L-1}\sum_{s=0}^{k}\sum_{x_s\in X_s} \mathbb{1}_t\{x_s\}\sqrt{\frac{2|\Omega|\ln(T|X|/\delta)}{\max\{1,N_t(x_s)\}}} + 2L|X|\sqrt{2T\ln\left(\frac{L}{\delta}\right)}$$

$$\leq 2L\sqrt{2L|X||\Omega|T\ln\left(\frac{T|X|}{\delta}\right)} + 2L|X|\sqrt{2T\ln\left(\frac{L}{\delta}\right)},$$

with probability at least $1-2\delta$, given the union bound over the Azuma-Hoeffding and the bound on the outcomes. Finally, with a union bound between the bound on the transitions and the outcomes (which are both encompassed by the clean event) and the Azuma-Hoeffding inequalities, with probability at least $1 - 4\delta$, we have:

$$\sum_{t\in[T]}\|q_t-\widehat{q}_t\|_1 \leq \mathcal{O}\left(L\sqrt{L|X||\Omega|T\ln\left(\frac{T|X|}{\delta}\right)} + L|X|\sqrt{T\ln\left(\frac{L}{\delta}\right)}+\right.$$

$$\left. + L^2|X|\sqrt{T|A||\Omega|\ln\left(\frac{T|X||\Omega||A|}{\delta}\right)} + L^2|X|\sqrt{T\ln\left(\frac{L}{\delta}\right)}\right)$$

$$\leq \mathcal{O}\left(L^2|X|\sqrt{T|A||\Omega|\ln\left(\frac{T|X||\Omega||A|}{\delta}\right)}\right),$$

which concludes the proof. □

## D.2 Regret

In the following section we show that Algorithm 2 attains $\tilde{\mathcal{O}}(\sqrt{T})$ regret. This is done showing that the confidence intervals over transitions, outcomes and sender reward concentrate at a rate of $\tilde{\mathcal{O}}(1/\sqrt{T})$.

**Theorem 1.** *Given any* $\delta \in (0,1)$, *with probability at least* $1 - 7\delta$, *Algorithm 2 attains regret* $R_T \leq \widetilde{\mathcal{O}}\left(L^2|X|\sqrt{T|A||\Omega|\ln\left(1/\delta\right)}\right)$.

*Proof.* We notice that the regret can be decomposed as follows:

$$R_T = \sum_{t\in[T]} r_S^\top(q^*-q_t) = \sum_{t\in[T]} r_S^\top(q^*-\widehat{q}_t) + \sum_{t\in[T]} r_S^\top(\widehat{q}_t-q_t).$$

The second term is bounded by Hölder inequality and applying Lemma 3. To bound the first term we notice that, under the clean event, and by definition of the linear program solved by Algorithm 2, it holds:

$$(r_S + 2\xi_{S,t})^\top \widehat{q}_t \geq (\overline{r}_{S,t} + \xi_{S,t})^\top \widehat{q}_t \geq (\overline{r}_{S,t} + \xi_{S,t})^\top q^* \geq r_S^\top q^*.$$

Thus, we have,

$$\sum_{t\in[T]} r_S^\top(q^*-\widehat{q}_t) \leq 2\sum_{t\in[T]} \xi_{S,t}^\top\widehat{q}_t = 2\sum_{t\in[T]} \xi_{S,t}^\top q_t + 2\sum_{t\in[T]} \xi_{S,t}^\top(\widehat{q}_t-q_t).$$

The second term is bounded by Hölder inequality and applying Lemma 3, which holds under the clean event, with probability at least $1 - 2\delta$. To bound the first term we employ Lemma 10 which holds under the clean event, with probability at least $1 - \delta$, and a union bound, which concludes the proof. □

## D.3 Violations

In the following section we show that Algorithm 2 attains $\tilde{\mathcal{O}}(\sqrt{T})$ violations. This is possible since, in the *full-feedback* setting, the incentive compatibility constraints collapse to standard linear constraints.

**Theorem 2.** *Given $\delta \in (0,1)$, with probability at least $1 - 7\delta$, Algorithm 2 attains violation $V_T \leq \widetilde{\mathcal{O}}\left(L^2|X|\sqrt{T|A||\Omega|\ln(1/\delta)}\right)$.*

*Proof.* In the proof, we compactly denote the receivers' best response in a given state-action pair $(x,a) \in X \times A$ at time $t \in [T]$ as $b^t(a,x) := b^{\phi^{\widehat{q}t}}(a,x)$. Furthermore, by employing the definition of the linear program and summing over $(x,\omega,a)$, for any episode $t$, under the clean event, it holds:

$$\sum_{x\in X, \omega\in\Omega, a\in A} \widehat{q}_t(x,\omega,a)\left(\overline{r}_{R,t}(x,\omega,a)+\xi_{R,t}(x,\omega,a)-\overline{r}_{R,t}(x,\omega,b^t(a,x))+\xi_{R,t}(x,\omega,b^t(a,x))\right) \geq 0,$$

which, in turn, implies that:

$$\sum_{x\in X, \omega\in\Omega, a\in A} \widehat{q}_t(x,\omega,a)\left(r_R(x,\omega,a)+2\xi_{R,t}(x,\omega,a)-r_R(x,\omega,b^t(a,x))+2\xi_{R,t}(x,\omega,b^t(a,x))\right) \geq 0.$$

Thus, noticing that, in the *full-feedback* setting, we have $\xi_{R,t}(x,\omega,a) = \xi_{R,t}(x,\omega,b^t(a,x))$, we obtain:

$$\sum_{x\in X, \omega\in\Omega, a\in A} \widehat{q}_t(x,\omega,a)\left(r_R(x,\omega,b^t(a,x)) - r_R(x,\omega,a)\right) \leq 4\sum_{x\in X, \omega\in\Omega, a\in A} \widehat{q}_t(x,\omega,a)\xi_{R,t}(x,\omega,a)$$

$$\leq 4\sum_{x\in X, \omega\in\Omega} \widehat{q}_t(x,\omega)\xi_{R,t}(x,\omega),$$

where $\xi_{R,t}(x,\omega) = \sqrt{\frac{\ln(3T|X||\Omega|/\delta)}{\max\{1, N_t(x,\omega)\}}}$.

Now we combine the previous equations to bound the first term of the last inequality as follows:

$$\sum_{t\in[T]}\sum_{x\in X, \omega\in\Omega, a\in A} \widehat{q}_t(x,\omega,a)\left(r_R(x,\omega,b^t(a,x)) - r_R(x,\omega,a)\right) \tag{4}$$

$$\leq 4\sum_{t\in[T]}\sum_{x\in X, \omega\in\Omega} \widehat{q}_t(x,\omega)\xi_{R,t}(x,\omega)$$

$$= 4\sum_{t\in[T]}\sum_{x\in X, \omega\in\Omega} q_t(x,\omega)\xi_{R,t}(x,\omega) + 4\sum_{t\in[T]}\sum_{x\in X, \omega\in\Omega} (\widehat{q}_t(x,\omega) - q_t(x,\omega))\xi_{R,t}(x,\omega)$$

$$\leq 4\sum_{t\in[T]}\sum_{x\in X, \omega\in\Omega} q_t(x,\omega)\xi_{R,t}(x,\omega) + \mathcal{O}\left(L^2|X|\sqrt{T|A||\Omega|\ln\left(\frac{T|X||\Omega||A|}{\delta}\right)}\right) \tag{5}$$

$$= 4\sum_{t\in[T]}\sum_{x\in X, \omega\in\Omega} \mathbb{1}_t\{x,\omega\}\xi_{R,t}(x,\omega) + 4\sum_{t\in[T]}\sum_{x\in X, \omega\in\Omega} (q_t(x,\omega) - \mathbb{1}_t\{x,\omega\})$$

$$+ \mathcal{O}\left(L^2|X|\sqrt{T|A||\Omega|\ln\left(\frac{T|X||\Omega||A|}{\delta}\right)}\right)$$

$$= 4\sum_{t\in[T]}\sum_{x\in X, \omega\in\Omega} \mathbb{1}_t(x,\omega)\xi_{R,t}(x,\omega) + 4\sum_{t\in[T]}\sum_{x\in X} (q_t(x) - \mathbb{1}_t(x))$$

$$+ \mathcal{O}\left(L^2|X|\sqrt{T|A||\Omega|\ln\left(\frac{T|X||\Omega||A|}{\delta}\right)}\right)$$

$$\leq 4\sum_{t\in[T]}\sum_{x\in X, \omega\in\Omega} \mathbb{1}_t(x,\omega)\xi_{R,t}(x,\omega) + 4|X|\sqrt{2T\ln\frac{X}{\delta}}$$

$$+ \mathcal{O}\left(L^2|X|\sqrt{T|A||\Omega|\ln\left(\frac{T|X||\Omega||A|}{\delta}\right)}\right) \tag{6}$$

$$\leq \sqrt{9L|X||\Omega|T\ln\frac{3T|X||\Omega|}{\delta}} + O\left(L^2|X|\sqrt{T|A||\Omega|\ln\left(\frac{T|X||\Omega||A|}{\delta}\right)}\right) \tag{7}$$

$$\leq O\left(L^2|X|\sqrt{T|A||\Omega|\ln\left(\frac{T|X||\Omega||A|}{\delta}\right)}\right),$$

where Inequality (5) holds by Hölder inequality and Lemma 3, which holds under the clean event, with probability at least $1 - 2\delta$, Inequality (6) follows by Azuma-Hoeffding and Inequality (7) by Cauchy-Schwarz inequality and observing that $1 + \sum_{t\in[T]}\frac{1}{\sqrt{t}} \leq 3\sqrt{T}$.

Finally, returning to the quantity of interest, we bound the cumulative violations as follows:

$$V_T := \sum_{t\in[T]}\sum_{x\in X, \omega\in\Omega, a\in A} q_t(x,\omega,a)\left(r_R(x,\omega,b^t(a,x)) - r_R(x,\omega,a)\right)$$

$$= \sum_{t\in[T]}\sum_{x\in X, \omega\in\Omega, a\in A} \widehat{q}_t(x,\omega,a)\left(r_R(x,\omega,b^t(a,x)) - r_R(x,\omega,a)\right)$$

$$+ \sum_{t\in[T]}\sum_{x\in X, \omega\in\Omega, a\in A} (q_t(x,\omega,a) - \widehat{q}_t(x,\omega,a))\left(r_R(x,\omega,b^t(a,x)) - r_R(x,\omega,a)\right)$$

$$\leq \sum_{t\in[T]}\sum_{x\in X, \omega\in\Omega, a\in A} \widehat{q}_t(x,\omega,a)\left(r_R(x,\omega,b^t(a,x)) - r_R(x,\omega,a)\right) + \sum_{t\in[T]}\sum_{x\in X, \omega\in\Omega, a\in A} |q_t(x,\omega,a) - \widehat{q}_t(x,\omega,a)|$$

$$\leq \sum_{t\in[T]}\sum_{x\in X, \omega\in\Omega, a\in A} \widehat{q}_t(x,\omega,a)\left(r_R(x,\omega,b^t(a,x)) - r_R(x,\omega,a)\right)$$

$$+ O\left(L^2|X|\sqrt{T|A||\Omega|\ln\left(\frac{T|X||\Omega||A|}{\delta}\right)}\right)$$

$$\leq O\left(L^2|X|\sqrt{T|A||\Omega|\ln\left(\frac{T|X||\Omega||A|}{\delta}\right)}\right),$$

where the last steps hold by Hölder inequality, Lemma 3 and the previous bound on the estimated occupancy measure. The final result holds with probability at least $1 - 7\delta$ employing a union bound over the clean event, which holds with probability at least $1 - 4\delta$, the Azuma-Hoeffding inequality used above, which holds with probability at least $1 - \delta$ and Lemma 3, which, under the clean event, holds with probability at least $1 - 2\delta$. $\qquad\square$

### D.4 Lower Bound

**Theorem 3.** *Let $\delta \in (0, 1/4)$. For every algorithm that guarantees $R_T \leq \mathcal{O}(\sqrt{T})$ with probability at least $1 - \delta$, there exists an absolute constant $\psi > 0$ and a problem instance in which the algorithm must have $V_T \geq \Omega(\sqrt{T})$ with probability larger than $\psi$.*

*Proof.* We consider two instances with a single possible outcome and a single state. In the following, we omit the dependence on the sender and receiver utility from these parameters. We assume that the sender's utility in the first instance is a deterministic function given by $r_S^1(a_1) = 1$ and $r_S^1(a_2) = 0$, while the receiver's utility is given by $r_R^2(a_1) \sim \text{Be}(1/2 + \epsilon)$ and $r_R^2(a_2) \sim \text{Be}(1/2)$. Meanwhile, the sender's utility in the second instance is $r_S^2(a_1) = 1$ and $r_S^1(a_2) = 0$, while the follower's utility is equal to $r_R^2(a_1) \sim \text{Be}(1/2 + \epsilon)$ and $r_R^2(a_2) \sim \text{Be}(1/2 + 2\epsilon)$, for some $\epsilon \in (0, 1/2)$. Thus, the sender's regret in the first instance is given by:

$$R_T^1 = \sum_{t=1}^{T} \phi^t(a_2),$$

since the optimal signaling scheme is the one that always recommends action $a_1 \in \mathcal{A}$ in the first instance. In the following, we define $\mathbb{P}^1$ (respectively, $\mathbb{P}^2$) as the probability measure induced by recommending, at each round, signaling schemes according to some algorithm in the first (respectively, second) instance. Then, we bound the probability that the regret in the first instance is larger than a constant $C \in \mathbb{N}$ as follows:

$$\mathbb{P}^1\left(R_T^1 \leq C\sqrt{T}\right) = \mathbb{P}^1\left(\sum_{t=1}^{T}\phi^t(a_2) \leq C\sqrt{T}\right) \geq 1 - \eta, \tag{8}$$

for some $\eta \in (0, 3/4)$. Furthermore, by Pinsker's inequality and Equation (8) the following holds.

$$\mathbb{P}^2 \left( \sum_{t=1}^{T} \phi^t(a_2) \leq C\sqrt{T} \right) \geq 1 - \eta - \sqrt{D_{KL}(\mathbb{P}^1, \mathbb{P}^2)}, \qquad (9)$$

where we denote with $D_{KL}(\cdot, \cdot)$ the Kullback-Leibler divergence between two probability measure. By means of the well known divergence decomposition, we have:

$$D_{KL}(\mathbb{P}^1, \mathbb{P}^2) \leq T D_{KL}(\text{Be}(1/2 + 2\epsilon), \text{Be}(1/2)) \leq 16\epsilon^2 T, \qquad (10)$$

where in the latter inequality we used the well known property ensuring that $D_{KL}(\text{Be}(p), \text{Be}(q)) \leq \frac{(p-q)^2}{q(1-q)}$. Furthermore, by means of the latter inequality and Equation (10) we have:

$$\mathbb{P}^2 \left( \sum_{t=1}^{T} \phi^t(a_2) \leq C\sqrt{T} \right) \geq 1 - \eta - \sqrt{16\epsilon^2 T}$$

We now consider the receiver's violations in the second instance which can be computed as follows:

$$V_T^2 = \sum_{t=1}^{T} \phi^t(a_1) \left( \bar{r}_R^2(a_2) - \bar{r}_R^2(a_1) \right) = \epsilon \sum_{t=1}^{T} \phi^t(a_1).$$

Then, by means of Equation (9) we get:

$$\mathbb{P}^2 \left( V_T^2 \geq \epsilon(T - C\sqrt{T}) \right) = \mathbb{P}^2 \left( \epsilon \sum_{t=1}^{T} \phi^t(a_1) \geq \epsilon(T - C\sqrt{T}) \right)$$

$$= \mathbb{P}^2 \left( T - \sum_{t=1}^{T} \phi^t(a_2) \geq T - C\sqrt{T} \right)$$

$$= \mathbb{P}^2 \left( \sum_{t=1}^{T} \phi^t(a_2) \leq C\sqrt{T} \right) \geq 1 - \eta - \sqrt{16\epsilon^2 T}.$$

Finally, by setting $\epsilon = \frac{1}{16\sqrt{T}}$ we get:

$$\mathbb{P}^1 \left( R_T^1 \leq C\sqrt{T} \right) \geq 1 - \eta$$

Then, the latter result implies that:

$$\mathbb{P}^2 \left( V_T^2 \geq \epsilon(T - C\sqrt{T}) \right) = \mathbb{P}^2 \left( V_T^2 \geq \Omega(\sqrt{T}) \right) \geq 1 - \eta - \sqrt{16\epsilon^2 T}$$

$$\geq \frac{3}{4} - \eta,$$

which concludes the proof. $\qquad \square$

# E  Partial feedback

## E.1  Regret

**Lemma 10.** *Under the event $\mathcal{E}(\delta)$, with probability at least $1 - \delta$, it holds:*

$$\sum_{t \in [T]} \xi_{S,t}^{\top} q_t \leq \mathcal{O} \left( \sqrt{L|X||\Omega||A|T \ln \left( \frac{T|X||\Omega||A|}{\delta} \right)} \right)$$

$$\sum_{t \in [T]} \xi_{R,t}^{\top} q_t \leq \mathcal{O} \left( \sqrt{L|X||\Omega||A|T \ln \left( \frac{T|X||\Omega||A|}{\delta} \right)} \right)$$

*Proof.* We bound the quantity of interest as follows:

$$\sum_{t\in[T]} \xi_{S,t}^{\top} q_t \leq \sum_{t\in[T]} \sum_{x\in X, \omega\in\Omega, a\in A} \xi_{S,t}(x,\omega,a)\mathbb{1}_t\{x,\omega,a\} + L\sqrt{2T\ln\frac{1}{\delta}} \tag{11}$$

$$= \sum_{t\in[T]} \sum_{x\in X, \omega\in\Omega, a\in A} \sqrt{\frac{\ln(3T|X||\Omega||A|/\delta)}{\max\{1, N_t(x,\omega,a)\}}}\mathbb{1}_t\{x,\omega,a\} + L\sqrt{2T\ln\frac{1}{\delta}}$$

$$\leq \sqrt{9\ln\left(\frac{3T|X||\Omega||A|}{\delta}\right)} \sum_{x\in X, \omega\in\Omega, a\in A} \sqrt{N_T(x,\omega,a)} + L\sqrt{2T\ln\frac{1}{\delta}} \tag{12}$$

$$\leq \sqrt{9\ln\left(\frac{3T|X||\Omega||A|}{\delta}\right)} \sqrt{|X||\Omega||A| \sum_{x\in X, \omega\in\Omega, a\in A} N_T(x,\omega,a)} + L\sqrt{2T\ln\frac{1}{\delta}} \tag{13}$$

$$\leq \sqrt{9L|X||\Omega||A|T\ln\left(\frac{3T|X||\Omega||A|}{\delta}\right)} + L\sqrt{2T\ln\frac{1}{\delta}}, \tag{14}$$

where Inequality (11) holds by the Azuma-Hoeffding inequality with probability $1-\delta$, Inequality (12) follows by observing that $1 + \sum_{t\in[T]} \frac{1}{\sqrt{t}} \leq 3\sqrt{T}$, Inequality (13) follows from the Cauchy-Schwarz inequality, and Inequality (14) holds, noticing that $\sum_{x\in X, \omega\in\Omega, a\in A} N_T(x,\omega,a) \leq LT$. With the same analysis, we can prove that the same upper bound holds for $\sum_{t\in[T]} \xi_{R,t}^{\top} q_t$, concluding the proof. $\qquad\square$

**Theorem 4.** *Given any $\delta \in (0,1)$, with probability at least $1 - 7\delta$, Algorithm 3 attains regret $R_T \leq \widetilde{\mathcal{O}}\left(NL|X||\Omega||A| + L^2|X|\sqrt{T|A||\Omega|\ln(1/\delta)}\right)$.*

*Proof.* As a first step, we decompose the sender's regret as follows:

$$R_T = \sum_{t\in[T]} r_S^{\top}(q^* - q_t)$$

$$= \sum_{t\in[T]} r_S^{\top}(q^* - \widehat{q}_t) + \sum_{t\in[T]} r_S^{\top}(\widehat{q}_t - q_t)$$

$$\leq \sum_{t\in[T]} r_S^{\top}(q^* - \widehat{q}_t) + \mathcal{O}\left(L^2|X|\sqrt{T|A||\Omega|\ln\left(\frac{T|X||\Omega||A|}{\delta}\right)}\right). \tag{15}$$

We observe that the last inequality holds under the event $\mathcal{E}(\delta)$, with a probability of at least $1 - 2\delta$, and it is derived by applying the Hölder inequality and employing Lemma 3. To bound the first term in Equation (15), we notice that under $\mathcal{E}(\delta)$, we have:

$$(r_S + 2\xi_{S,t})^{\top}\widehat{q}_t \geq (\overline{r}_{S,t} + \xi_{S,t})^{\top}\widehat{q}_t \geq (\overline{r}_{S,t} + \xi_{S,t})^{\top}q^* \geq r_S^{\top}q^*,$$

for each $t > N|X||\Omega||A|$ because of the optimality of $\widehat{q}_t$. Thus, rearranging the latter chain of inequalities we have:

$$\sum_{t\in[T]} r_S^{\top}(q^* - \widehat{q}_t) = \sum_{t\leq N|X||\Omega||A|} r_S^{\top}(q^* - \widehat{q}_t) + \sum_{t>N|X||\Omega||A|} r_S^{\top}(q^* - \widehat{q}_t)$$

$$\leq NL|X||\Omega||A| + 2\left(\sum_{t\in[T]} \xi_{S,t}^{\top}(\widehat{q}_t - q_t) + \sum_{t\in[T]} \xi_{S,t}^{\top}q_t\right)$$

$$\leq NL|X||\Omega||A| + \mathcal{O}\left(L^2|X|\sqrt{T|A||\Omega|\ln\left(\frac{T|X||\Omega||A|}{\delta}\right)}\right).$$

In the first inequality above, we employ the fact that the support of each reward function belongs to $[0, 1]$, while in the second inequality, we make use of Lemma 3, the Hölder inequality, and Lemma 10,

which hold with a probability of at least $1 - 3\delta$. Substituting the latter inequality into Equation (15), we obtain:

$$R_T \leq \mathcal{O}\left( NL|X||\Omega||A| + L^2|X|\sqrt{T|A||\Omega|\ln\left(\frac{T|X||\Omega||A|}{\delta}\right)} \right).$$

Finally, we observe that the previous upper bound holds with probability at least $1 - 7\delta$. This follows by employing a union bound and observing that $\mathcal{E}(\delta)$ holds with a probability at least $1 - 4\delta$, which concludes the proof. □

## E.2 Violations

In the following we denote the receivers' best response in a given action $a \in A$ and state $x \in X$ as $b^t(a, x) := b^{\phi^{\widehat{q}_t}}(a, x)$.

**Lemma 11.** *Under the event $\mathcal{E}(\delta)$ the following holds:*

$$V_T \leq \mathcal{O}\left( L^2|X|\sqrt{T|A||\Omega|\ln\left(\frac{T|X||\Omega||A|}{\delta}\right)} \right) + \sum_{t\in[T]}\sum_{x\in X, \omega\in\Omega, a\in A} q_t(x, \omega, a)\xi_{R,t}(x, \omega, b^t(a, x)),$$

*with probability at least $1 - 3\delta$.*

*Proof.* As a first step, we observe that by employing the definition of $\xi_{R,t}$ and noticing that $\widehat{q}_t$ is a feasible solution to LP (2) for each $t \in [T]$ under the event $\mathcal{E}(\delta)$, we have:

$$\sum_{x\in X, \omega\in\Omega, a\in A} \widehat{q}_t(x, \omega, a) \left( r_R(x, \omega, a) + 2\xi_{R,t}(x, \omega, a) - r_R(x, \omega, b^t(a, x)) + 2\xi_{R,t}(x, \omega, b^t(a, x)) \right) \geq$$

$$\sum_{x\in X, \omega\in\Omega, a\in A} \widehat{q}_t(x, \omega, a) \left( \overline{r}_{R,t}(x, \omega, a) + \xi_{R,t}(x, \omega, a) - \overline{r}_{R,t}(x, \omega, b^t(a, x)) + \xi_{R,t}(x, \omega, b^t(a, x)) \right) \geq 0.$$

Then, rearranging the above inequality we get:

$$\sum_{x\in X, \omega\in\Omega, a\in A} \widehat{q}_t(x, \omega, a) \left( r_{R,t}(x, \omega, b^t(a, x)) - r_{R,t}(x, \omega, a) \right)$$

$$\leq 2 \sum_{x\in X, \omega\in\Omega, a\in A} \widehat{q}_t(x, \omega, a) \left( \xi_{R,t}(x, \omega, a) + \xi_{R,t}(x, \omega, b^t(a, x)) \right). \tag{16}$$

Furthermore, we can decompose the receivers' violations as follows:

$$V_T = \sum_{t\in[T]}\sum_{x\in X, \omega\in\Omega, a\in A} (q_t(x, \omega, a) \pm \widehat{q}_t(x, \omega, a)) \left( r_R(x, \omega, b^t(a, x)) - r_R(x, \omega, a) \right)$$

$$\leq \mathcal{O}\left( L^2|X|\sqrt{T|A||\Omega|\ln\left(\frac{T|X||\Omega||A|}{\delta}\right)} \right) +$$

$$\sum_{t\in[T]}\sum_{x\in X, \omega\in\Omega, a\in A} \widehat{q}_t(x, \omega, a) \left( r_R(x, \omega, b^t(a, x)) - r_R(x, \omega, a) \right)$$

$$\leq \mathcal{O}\left( L^2|X|\sqrt{T|A||\Omega|\ln\left(\frac{T|X||\Omega||A|}{\delta}\right)} \right) +$$

$$\sum_{t\in[T]}\sum_{x\in X, \omega\in\Omega, a\in A} (\widehat{q}_t(x, \omega, a) \pm q_t(x, \omega, a)) \left( \xi_{R,t}(x, \omega, a) + \xi_{R,t}(x, \omega, b^t(a, x)) \right)$$

$$\leq \mathcal{O}\left( L^2|X|\sqrt{T|A||\Omega|\ln\left(\frac{T|X||\Omega||A|}{\delta}\right)} \right)$$

$$+ \sum_{t\in[T]}\sum_{x\in X, \omega\in\Omega, a\in A} q_t(x, \omega, a) \left( 2\xi_{R,t}(x, \omega, a) + 2\xi_{R,t}(x, \omega, b^t(a, x)) \right)$$

$$\leq \mathcal{O}\left( L^2|X|\sqrt{T|A||\Omega|\ln\left(\frac{T|X||\Omega||A|}{\delta}\right)}\right) + 2\sum_{t\in[T]}\sum_{x\in X,\omega\in\Omega,a\in A} q_t(x,\omega,a)\xi_{R,t}(x,\omega,b^t(a,x)),$$

where the first and third inequalities hold by Lemma 3, the second inequality is a consequence of Inequality (16), and the third inequality follows by means of Lemma 10, which holds with a probability of at least $1-\delta$. Therefore, employing a union bound over the events of Lemma 3 and Lemma 10, the previous result holds with probability at least $1-3\delta$, under the clean event. $\qquad\square$

**Theorem 5.** *Given any $\delta \in (0,1)$, with probability at least $1-9\delta$, Algorithm 3 attains cumulative violation $V_T \leq \widetilde{\mathcal{O}}\left(\rho\left(|A|\frac{T}{\sqrt{N}} + |A|\sqrt{N} + L^2\sqrt{T}\right)\right)$, where $\rho := (|X||\Omega||A|)^{3/2}\sqrt{\ln\left(1/\delta\right)}$.*

*Proof.* As a preliminary observation, we notice that Algorithm 3 is divided into $N$ epochs of length $\ell = |X||\Omega||A|$, where in each epoch, Algorithm 3 maximizes the probability of visiting each triplet $(x,\omega,a)$. In the following, we define $t_j(x,\omega,a) \in [T]$ as the round in which Algorithm 3 maximizes the occupancy of the triplet $(x,\omega,a)$ in the epoch $j \in [N-1]$. Formally:

$$t_j(x,\omega,a) := \{t \in [\,j\ell+1,\ldots,(j+1)\ell\,] \mid \sum_{x'\in X} q(x,\omega,a,x') \text{ is the objective function of Program (2) }\}$$

Furthermore, for each occupancy measure $q_t$ with $t \in [T]$, the following holds:

$$q_t(x,\omega,a) = q(x,\omega,b^t(a,x)) \leq q_{t_j(x,\omega,b^t(a,x))}(x,\omega,b^t(a,x)) \tag{17}$$

for each $j \in [N-1]$ where $q \in \mathcal{Q}$ is an occupancy measure that satisfies the IC constraints of the offline optimization problem (see Program (1)). The first equality above follows by observing that there always exists an occupancy that satisfies the IC constraints that recommends action $b^t(a,x) \in A$ instead of $a \in A$ in the state $x \in X$ with the same probability of $q_t$. The inequality, on the other hand, follows by observing that the space of occupancy measures satisfying the IC constraint of the offline optimization problem (1) is always a subset of the feasibility set of Program (2).

Furthermore, by Lemma 11 we have that:

$$V_T \leq \mathcal{O}\left( L^2|X|\sqrt{T|A||\Omega|\ln\left(\frac{T|X||\Omega||A|}{\delta}\right)}\right) + \sum_{t\in[T]}\sum_{x\in X,\omega\in\Omega,a\in A} q_t(x,\omega,a)\xi_{R,t}(x,\omega,b^t(a,x)),$$

We focus on bounding the second term in the inequality above in the first $N\ell$ rounds of Algorithm 3. Thus, with probability at least $1-\delta$ we have:

$$\sum_{t\leq N\ell}\sum_{x\in X,\omega\in\Omega,a\in A} q_t(x,\omega,a)\xi_{R,t}(x,\omega,b^t(a,x))$$

$$\leq \sum_{t=1}^{\ell}\sum_{x\in X,\omega\in\Omega,a\in A} q_t(x,\omega,a)\xi_{R,t}(x,\omega,b^t(a,x))+$$

$$+ \sum_{t=\ell+1}^{N\ell}\sum_{x\in X,\omega\in\Omega,a\in A} q_t(x,\omega,a)\left(\xi_{R,t}(x,\omega,b^t(a,x))\right)$$

$$\leq L|X||\Omega||A| + \sum_{x\in X,\omega\in\Omega,a\in A}\left(\sum_{j=1}^{N-1}\sum_{t=j\ell}^{(j+1)\ell} q_t(x,\omega,a)\xi_{R,t}(x,\omega,b^t(a,x))\right) \tag{18}$$

$$\leq L|X||\Omega||A| + \sum_{x\in X,\omega\in\Omega,a\in A}\left[\sum_{a'\in A}\left(\sum_{j=1}^{N-1}\sum_{t=j\ell}^{(j+1)\ell} q_t(x,\omega,a')\left(\xi_{R,t}(x,\omega,a')\mathbb{1}\{b^t(a,x)=a'\}\right)\right)\right]$$

$$\leq L|X||\Omega||A|+$$

$$+ \sum_{x\in X,\omega\in\Omega,a\in A}\left[\sum_{a'\in A}\left(\sum_{j=1}^{N-1} q_{t_j(x,\omega,a')}(x,\omega,a')\sum_{t=j\ell}^{(j+1)\ell}\left(\xi_{R,t}(x,\omega,a')\mathbb{1}\{b^t(a,x)=a'\}\right)\right)\right] \tag{19}$$

$$\leq L|X||\Omega||A| + \sum_{x\in X, \omega\in\Omega, a\in A}\left[\sum_{a'\in A}\left(\sum_{j=1}^{N-1} q_{t_j(x,\omega,a')}(x,\omega,a')\sum_{t=j\ell}^{(j+1)\ell}\xi_{R,t}(x,\omega,a')\right)\right] \quad (20)$$

$$\leq L|X||\Omega||A| + \sqrt{\ln\left(\frac{2T|X||\Omega||A|}{\delta}\right)}\cdot$$
$$\cdot\sum_{x\in X, \omega\in\Omega, a\in A}\left[\sum_{a'\in A}\left(\sum_{j=1}^{N-1} q_{t_j(x,\omega,a')}(x,\omega,a')\sum_{t=j\ell}^{(j+1)\ell}\frac{1}{\sqrt{\max\{1, N_t(x,\omega,a')\}}}\right)\right]$$

$$\leq L|X||\Omega||A| + \ell\sqrt{\ln\left(\frac{2T|X||\Omega||A|}{\delta}\right)}\cdot$$
$$\cdot\sum_{x\in X, \omega\in\Omega, a\in A}\left[\sum_{a'\in A}\left(\sum_{j=1}^{N-1}\frac{q_{t_j(x,\omega,a')}(x,\omega,a')}{\sqrt{\max\{1, N_j(x,\omega,a')\}}}\right)\right] \quad (21)$$

$$\leq L|X||\Omega||A| + \ell\sqrt{\ln\left(\frac{2T|X||\Omega||A|}{\delta}\right)}\cdot$$
$$\cdot\left(\sum_{x\in X, \omega\in\Omega, a\in A}\left[\sum_{a'\in A}\left(\sum_{j=1}^{N-1}\frac{\mathbb{1}_{t_j(x,\omega,a')}(x,\omega,a')}{\sqrt{\max\{1, N_j(x,\omega,a')\}}}\right)\right] + L|A|\sqrt{2N\ln\frac{1}{\delta}}\right) \quad (22)$$

$$\leq L|X||\Omega||A| + 3\ell\sqrt{\ln\left(\frac{2T|X||\Omega||A|}{\delta}\right)}\cdot$$
$$\cdot\left(\sum_{x\in X, \omega\in\Omega, a\in A}\left[\sum_{a'\in A}\sqrt{\sum_{i=1}^{N}\mathbb{1}_{t_i(x,\omega,a')}}\right] + L|A|\sqrt{2N\ln\frac{1}{\delta}}\right)$$

$$\leq L|X||\Omega||A| + 3\ell\sqrt{\ln\left(\frac{2T|X||\Omega||A|}{\delta}\right)}\cdot$$
$$\cdot\left(\sum_{x\in X, \omega\in\Omega, a\in A}\left[\sum_{a'\in A}\sqrt{N_{N\ell}(x,\omega,a')}\right] + L|A|\sqrt{2N\ln\frac{1}{\delta}}\right) \quad (23)$$

$$\leq L|X||\Omega||A| + 3\ell|A|\sqrt{\ln\left(\frac{2T|X||\Omega||A|}{\delta}\right)}\left(\sum_{x\in X, \omega\in\Omega, a'\in A}\sqrt{N_{N\ell}(x,\omega,a')} + L\sqrt{2N\ln\frac{1}{\delta}}\right)$$

$$\leq L|X||\Omega||A| + 3\ell|A|\sqrt{\ln\left(\frac{2T|X||\Omega||A|}{\delta}\right)}\left(\sqrt{LN\ell} + L\sqrt{2N\ln\frac{1}{\delta}}\right), \quad (24)$$

where we let $N_j(x,\omega,a) = \sum_{i\leq j}\mathbb{1}_{t_i(x,\omega,a)}(x,\omega,a)$ for the the sake of simplicity. Furthermore, we notice that Inequality (18) follows observing that $\xi_{r,t}(x,\omega,a) \leq 1$ for each $(x,\omega,a) \in X\times\Omega\times A$ and $t\in[T]$, and because the occupancy defines a probability distribution over each layer $k\in[0,\ldots,L]$. Inequality (19) holds thanks to Inequality (17). Inequality (20) follows because each indicator function takes value of at most one. Inequality (21) follows by observing that the number of times that the triplet $(x,\omega,a')$ is visited overall is always greater or equal to the the number of times such a triplet has been visited during the rounds in which Algorithm 3 maximizes the exploration of that triplet. Inequality (22) holds with probability at least $1-\delta$ and follows from the Azuma-Hoeffding inequality, and Inequality (24) holds, noticing that $\sum_{x\in X, \omega\in\Omega, a\in A} N_T(x,\omega,a) \leq LN\ell$ and employing the Cauchy-Schwarz inequality.

We focus on bounding the cumulative violations suffered in the remaining $T-N\ell$ rounds of Algorithm 3. With probability at least $1-\delta$ the following holds:

$$\sum_{t>N\ell}\sum_{x\in X, \omega\in\Omega, a\in A} q_t(x,\omega,a)\xi_{R,t}(x,\omega,b^t(a,x))$$

$$\leq \sum_{x\in X, \omega\in\Omega, a\in A} \left( \sum_{a'\in A} \sum_{t>N\ell} q_t(x,\omega,a')\xi_{R,t}(x,\omega,a')\mathbb{1}_t\{b^t(a,x)=a'\} \right)$$

$$\leq \sum_{x\in X, \omega\in\Omega, a\in A} \sqrt{\ln\left(\frac{2T|X||\Omega||A|}{\delta}\right)} \sum_{a'\in A} q_{t_N(x,\omega,a')}(x,\omega,a') \sum_{t>N\ell} \frac{1}{\sqrt{\max\{1, N_t(x,\omega,a')\}}} \quad (25)$$

$$\leq |A|\sqrt{\ln\left(\frac{2T|X||\Omega||A|}{\delta}\right)} \sum_{x\in X, \omega\in\Omega, a\in A} q_{t_N(x,\omega,a)}(x,\omega,a) \sum_{t>N\ell} \frac{1}{\sqrt{\max\{1, N_{N\ell}(x,\omega,a)\}}}$$

$$\leq |A|\sqrt{\ln\left(\frac{2T|X||\Omega||A|}{\delta}\right)} \sum_{x\in X, \omega\in\Omega, a\in A} q_{t_N(x,\omega,a)}(x,\omega,a) \frac{(T-N\ell)}{\sqrt{\max\{1, N_{N\ell}(x,\omega,a)\}}}$$

$$\leq |A|\sqrt{\ln\left(\frac{2T|X||\Omega||A|}{\delta}\right)} \sum_{x\in X, \omega\in\Omega, a\in A} \frac{N_{N\ell}(x,\omega,a)+L\sqrt{2N\ln\frac{1}{\delta}}}{N} \frac{(T-N\ell)}{\sqrt{\max\{1, N_{N\ell}(x,\omega,a)\}}} \quad (26)$$

$$\leq |A|\sqrt{\ln\left(\frac{2T|X||\Omega||A|}{\delta}\right)} \frac{T}{N}\left( \sqrt{LN\ell} + L\ell\sqrt{2N\ln\frac{1}{\delta}} \right) \quad (27)$$

$$\leq 2|A|\sqrt{\ln\left(\frac{2T|X||\Omega||A|}{\delta}\right)} \frac{T}{\sqrt{N}} L\ell\sqrt{2\ln\frac{1}{\delta}}. \quad (28)$$

Inequality (25) holds thanks to Inequality (17) and observing that the indicator function takes value of at most one. Inequality (27) holds, noticing that $\sum_{x\in X, \omega\in\Omega, a\in A} N_T(x,\omega,a) \leq LN\ell$ and employing the Cauchy-Schwarz inequality. Inequality (26) holds with probability at least $1-\delta$ and follows by employing the Azuma-Hoeffding and observing the following:

$$N_{N\ell}(x,\omega,a) \geq \sum_{k=1}^{N} \mathbb{1}_{t_k}(x,\omega,a)$$

$$\geq \sum_{k=1}^{N} q_{t_k}(x,\omega,a) - L\sqrt{2N\ln\frac{1}{\delta}}$$

$$\geq N q_{t_N(x,\omega,a)}(x,\omega,a) - L\sqrt{2N\ln\frac{1}{\delta}},$$

which can be written as follows:

$$\frac{N_{N\ell}(x,\omega,a)+L\sqrt{2N\ln\frac{1}{\delta}}}{N} \geq q_{t_N(x,\omega,a)}(x,\omega,a).$$

Finally, thanks to Lemma 11 and employing Inequality (24) and Inequality (27) we get:

$$V_T \leq \widetilde{\mathcal{O}}\left( \rho\left( |A|\frac{T}{\sqrt{N}} + |A|\sqrt{N} + L^2\sqrt{T} \right) \right).$$

With $\rho := (|X||\Omega||A|)^{3/2}\sqrt{\ln(1/\delta)}$, such a result holds with a probability of at least $1-9\delta$, employing a union bound and observing that $\mathcal{E}(\delta)$ holds with a probability of at least $1-4\delta$, Lemma 11 holds with a probability of at least $1-3\delta$, and both Inequality (24) and Inequality (27) hold with a probability of at least $1-\delta$. □

### E.3 Lower bound

**Theorem 6.** *Given $\alpha \in [1/2, 1]$, there is no learning algorithm achieving both $R_T = o(T^\alpha)$ and $V_T = o(T^{1-\alpha/2})$ with probability greater or equal to a fixed constant $\psi > 0$.*

*Proof.* We consider two instances with a single possible outcome and a single state. In the following, we omit the dependence on the sender and receiver utility from these parameters. We assume that the

sender's utility in the first instance is a deterministic function given by $r_S^1(a_1) = 1$ and $r_S^1(a_2) = 0$, while the receiver's utility is given by $r_R^2(a_1) \sim \text{Be}(1/2 + \epsilon)$ and $r_R^2(a_2) \sim \text{Be}(1/2)$. Meanwhile, the sender's utility in the second instance is $r_S^2(a_1) = 1$ and $r_S^1(a_2) = 0$, while the follower's utility is equal to $r_R^2(a_1) \sim \text{Be}(1/2 + \epsilon)$ and $r_R^2(a_2) \sim \text{Be}(1/2 + 2\epsilon)$, for some $\epsilon \in (0, 1/2)$. Thus, the sender's regret in the first instance is given by:

$$R_T^1 = \sum_{t=1}^{T} \phi^t(a_2),$$

since the optimal signaling scheme is the one that always recommends action $a_1 \in \mathcal{A}$ in the first instance. In the following, we define $\mathbb{P}^1$ (respectively, $\mathbb{P}^2$) as the probability measure induced by recommending, at each round, signaling schemes according to some algorithm in the first (respectively, second) instance. Then, we bound the probability that the regret in the first instance is larger than a constant $C \in \mathbb{N}$ as follows:

$$\mathbb{P}^1\left(R_T^1 \leq C\right) = \mathbb{P}^1\left(\sum_{t=1}^{T} \phi^t(a_2) \leq C\right) \geq 1 - \eta, \tag{29}$$

for some $\eta \in (0, 1)$. Furthermore, by Pinsker's inequality and Equation (29) the following holds.

$$\mathbb{P}^2\left(\sum_{t=1}^{T} \phi^t(a_2) \leq C\right) \geq 1 - \eta - \sqrt{D_{KL}(\mathbb{P}^1, \mathbb{P}^2)}, \tag{30}$$

where we denote with $D_{KL}(\cdot, \cdot)$ the Kullback-Leibler divergence between two probability measure. By means of the well known divergence decomposition, we have:

$$D_{KL}(\mathbb{P}^1, \mathbb{P}^2) \leq \mathbb{E}^1\left[\sum_{t=1}^{T} \phi^t(a_2)\right] D_{KL}(\text{Be}(1/2 + 2\epsilon), \text{Be}(1/2)) \leq 16\epsilon^2 \mathbb{E}^1\left[\sum_{t=1}^{T} \phi^t(a_2)\right], \tag{31}$$

where in the latter inequality we used the well known property ensuring that $D_{KL}(\text{Be}(p), \text{Be}(q)) \leq \frac{(p-q)^2}{q(1-q)}$. Then, by reverse Markov inequality and Equation (29) we get:

$$\mathbb{E}^1\left[\sum_{t=1}^{T} \phi^t(a_2)\right] \leq \mathbb{P}^1\left(\sum_{t=1}^{T} \phi^t(a_2) \geq C\right)(T - C) + C \leq \eta(T - C) + C,$$

Furthermore, by means of the latter inequality and Equation (31) we have:

$$\mathbb{P}^2\left(\sum_{t=1}^{T} \phi^t(a_2) \leq C\right) \geq 1 - \eta - \sqrt{16\epsilon^2(\eta(T - C) + C)}$$

We now consider the receiver's violations in the second instance which can be computed as follows:

$$V_T^2 = \sum_{t=1}^{T} \phi^t(a_1)\left(\bar{r}_R^2(a_2) - \bar{r}_R^2(a_1)\right) = \epsilon \sum_{t=1}^{T} \phi^t(a_1).$$

Then, by means of Equation (30) we get:

$$\mathbb{P}^2\left(V_T^2 \geq \epsilon(T - C)\right) = \mathbb{P}^2\left(\epsilon \sum_{t=1}^{T} \phi^t(a_1) \geq \epsilon(T - C)\right)$$

$$= \mathbb{P}^2\left(T - \sum_{t=1}^{T} \phi^t(a_2) \geq T - C\right)$$

$$= \mathbb{P}^2\left(\sum_{t=1}^{T} \phi^t(a_2) \leq C\right) \geq 1 - \eta - \sqrt{16\epsilon^2(\eta(T - C) + C)}.$$

Finally, by setting $C = \frac{T^\alpha}{2}$ and $\epsilon = \frac{T^{-\alpha/2}}{16}$ and $\eta = \frac{T^{\alpha-1}}{2}$ we get:

$$\mathbb{P}^1\left(R_T^1 \leq C\right) \geq 1 - \eta$$

$$\mathbb{P}^1\left(R_T^1 \le \frac{T^\alpha}{2}\right) \ge 1 - \frac{T^{\alpha-1}}{2} \ge \frac{1}{2},$$

since $\alpha \in [1/2, 1]$. Then, the latter result implies that:

$$\mathbb{P}^2\left(V_T^2 \ge \frac{1}{32}T^{1-\alpha/2}\right) \ge \mathbb{P}^2\left(V_T^2 \ge \epsilon(T-C)\right) \ge 1 - \eta - \sqrt{16\epsilon^2(\eta(T-C)+C)}$$

$$\ge 1 - \frac{T^{\alpha-1}}{2} - \sqrt{\frac{T^{-\alpha}}{16}\left(\frac{T^\alpha}{2} + \frac{T^\alpha}{2}\right)} \ge \frac{1}{4},$$

which concludes the proof. $\qquad\square$

