# OpenReview forum: "Markov Persuasion Processes: Learning to Persuade From Scratch"
_NeurIPS.cc/2025/Conference — NeurIPS 2025 poster_

### Official Review · Reviewer_owP8 · 2025-07-03

**Clarity:** 3
**Significance:** 3
**Originality:** 3
**Rating:** 4
**Confidence:** 3

**Summary:**

The paper tackles Markov persuasion processes (MPP) where a sender interacts with a sequence of myopic receivers, where the sender must learn how to disclose information optimally without access to the environment’s dynamics or the preferences of he receivers. It proposes a learning algorithm for the sender for the case of partial feedback where the regret (w.r.t. an policy optimal of information disclosure) is sublinear w.r.t. the episodes and also the violations of the persuasiveness constraints remain sublinear. The paper also shows that the sublinear bounds it obtains in the analysis are tight.

**Questions:**

- Can the authors comment on practical applications of their setup and proposed algorithm?

- Can the authors comment on the practical efficiency of their algorithm? Some, even if of small-size, computaitonal results would better emphasize the relevance of the work.

- Can the authors comments on the numerics of their algorithm? I am asking about the many linear programming problems that the proposed algorithm solves. Are they well-conditioned or can there be cases (e.g, with small state spaces, symmetric transitions, or almost identical receivers) where the persuasiveness and occupancy constraints become very similar, leading to rank-deficient basis matrices?

**Ethical Concerns:**

["NO or VERY MINOR ethics concerns only"]

**Final Justification:**

The authors provided sensible responses to my questions.

**Limitations:**

Yes (there was nothing to report).

**Paper Formatting Concerns:**

All good.

**Quality:**

3

**Strengths And Weaknesses:**

The paper is well written and the relevance of its results clearly explained -- overall, the paper looks rather impressive when it comes to its results. Due to its technical nature, though, most of the paper's results are confined to the appendices,, and not much is done in the actual paper to summarize how the results of its many theorems have been obtained. The paper is also completely theoretical in nature, and lacks any empirical validation. As far as I can see, it's the first learning framework for MPPs where the sender does not know the transition dynamics, the priors, or the utility functions; as such, I think a bit of extra effort should be made in explaining in some detail what applications may benefit from this framework (some applications are mentioned, but only in a very abstract way). Also, if the motivations of the paper are not purely theoretical, some form of empirical validation (which is lacking) should be included.

---

> ### Author Rebuttal · Authors · 2025-07-30
>
> We thank the Reviewer for the positive evaluation of our work.
>
> Re: *"Can the authors comment on practical applications of their setup and proposed algorithm?"*
>
> In the following, we provide a simple illustrative example. Consider a product recommendation platform that suggests items to multiple users. In this setting, the platform acts as the sender, while the users are the receivers who select actions, such as clicking on or purchasing recommended products. The model distinguishes between two types of information: the outcome ($\omega$), which represents the sender’s private knowledge (e.g., product features and availability), and the state ($s$), which captures the dynamic environment (e.g., user browsing history). Outcomes are drawn from a prior distribution and are used by the platform to personalize its suggestions. The state evolves over time based on the user’s action, the previous state, and the realized outcome. Both the sender’s and the receivers’ utilities depend on the chosen action, the current state, and the realized outcome.
>
> We will surely include this discussion in the final version of the paper.
>
> Re: *Can the authors comment on the practical efficiency of their algorithm? Some, even if of small-size, computaitonal results would better emphasize the relevance of the work.*
>
> Our algorithm requires solving a linear program at each round $t \in [T]$, with a number of variables and constraints that is polynomial in $|X|$, $|A|$, and $|\Omega|$; thus, the per-round update can be solved in polynomial time and the running time depends on the employed solver. Current solvers can solve these kinds of problems efficiently. The main goal of this work is theoretical, but we leave as an interesting direction for future work the evaluation of our algorithms' performance in practice.
>
>
>
> Re: *"Can the authors comments on the numerics of their algorithm? I am asking about the many linear programming problems that the proposed algorithm solves. Are they well-conditioned or can there be cases (e.g, with small state spaces, symmetric transitions, or almost identical receivers) where the persuasiveness and occupancy constraints become very similar, leading to rank-deficient basis matrices?"*
>
> From a theoretical point of view, any linear program can be solved in polynomial time. Clearly, since our model is fully general, these kind of problems could occur. As we mentioned above, we leave to future works the study of more applicative-oriented settings where these kinds of questions have a clearer answer.

---

> > ### Comment · Reviewer_owP8 · 2025-08-05
> >
> > Thanks for the reply. I concur that adding a practical example to the paper is rather important. Regarding the polynomiality aspect, I should say that this
> >
> > "any linear program can be solved in polynomial time"
> >
> > is not technically true. LPs are solved in polytime only w.r.t. (besides the number of variables and constraints) the bit-length of the input coefficients---i.e., linear programming is only weakly polynomially solvable. Hence my questions about the numerics of the algorithm.

---

> > > ### Author Response · Authors · 2025-08-06
> > >
> > > We thank the Reviewer for pointing out this aspect. All the quantities that define the matrices of the LP (Problem 2), which is solved at each round, either have the same bit complexity as the parameters of the MPP or are obtained through nonlinear operations (such as logarithms or square roots) applied to them. This is the case, for instance, for the quantities $\xi_R$, $\xi_S$. In such cases, rather than using the exact value, we may round up to the nearest rational number with a suitable bit complexity, say $B$. The LP can then be solved in time polynomial in $B$ (and in the bit complexity of the other parameters of the instance), and the error introduced in the final regret bound is exponentially small in $B$, i.e., of order $2^{-B} \cdot T$. By choosing $B \propto \log(T)$, the regret contribution of this component remains constant and thus negligible.
> > >
> > > Finally, let us remark that, in the online learning literature, all the issues related to numerical precision and bit complexity are not usually the main focus of the analysis (as they do not usually affect it), which is instead the derivation of sublinear regret guarantees.

---

> > > > ### Comment · Reviewer_owP8 · 2025-08-08
> > > >
> > > > Thanks for the reply. I'm satisfied with the response.

---

### Official Review · Reviewer_ocM5 · 2025-07-03

**Clarity:** 3
**Significance:** 3
**Originality:** 3
**Rating:** 4
**Confidence:** 4

**Summary:**

The paper considers the no-regret learning problem in Markov persuasion processes (MPPs), relaxing the assumption of previous work that the agent knows the receiver's utility function. It introduces the Optimistic Persuasive Policy Search (OPPS) algorithm, capable of learning the receiver's utility function from scratch, and provides rigorous regret bounds for the proposed algorithm under both full and partial feedback settings. The paper also provide a lower bound analysis on the optimality of the regret achieved by OPPS.

**Questions:**

Can the authors explicitly point out the improvement of their paper since the last version?

Overall, I believe my technical questions about this paper have been answered again and again, and I have become overall satisfied with their answers, given that many problems cannot be meaningfully addressed under the current framework. I think the authors could have used some of their answers to acknowledge the limitations of this work.

**Ethical Concerns:**

["NO or VERY MINOR ethics concerns only"]

**Final Justification:**

I have reviewed this paper for several times in different venue. Its reviews were always around the borderline and I think the paper deserves a chance to be finally accepted.

**Limitations:**

No, though I see that there is no potential negative societal impact from this theory work, I think the authors could have better acknowledge the limitation of their modeling assumptions or technical results.

**Quality:**

3

**Strengths And Weaknesses:**

Strength

- The authors consider the RL problem in MPPs, which is an interesting new problem from prior work. The analysis of this paper relies on the techniques of concentration measure, which offers a novel perspective to the studies of MPPs.

- The overall writing of this paper is clear, and it is clear to me how their results can be derived from the technical lemmas and proof sketches outlined in the main body.

Weakness

- I have reviewed earlier versions of this paper at several different venues before. The outcome of this paper has been at the borderline for acceptance and I have not noticed any big change since then. I support the authors' decision for resubmission but I notice there's almost no additional effort made to improve the paper.

- I am not sure if the limitations of this paper have been clearly acknowledged by the authors in Section 2.1 and 2.2.

- The paper measures the learning performance with two quantities, total regret and total violation of persuasiveness. In contrast, [Wu et al. 2022] ensures persuasiveness in high probability and only considers the total regret. I can see the technical reasons behind this choice, but is there any practical motivation for this evaluation metric? Why would the receiver follow the persuasion if the recommendation is not completely incentive-compatible? Are receivers no-regret learners as well so they can tolerate \sqrt{T} loss? Is it possible to convert any result of \sqrt{T} in total regret and total violation of persuasiveness into an equivalent regret bound where the persuasiveness is guaranteed with high probability by introducing certain robustification/pessimism design to the signaling scheme? Without these explanations, I am not quite convinced that this learning result in MPPs is meaningful enough.

---

> ### Author Rebuttal · Authors · 2025-07-30
>
> We thank the Reviewer for the positive evaluation of our work.
>
> Re: *"Can the authors explicitly point out the improvement of their paper since the last version? Overall, I believe my technical questions about this paper have been answered again and again, and I have become overall satisfied with their answers, given that many problems cannot be meaningfully addressed under the current framework. I think the authors could have used some of their answers to acknowledge the limitations of this work."*
>
> Compared to the initial version of the paper, we have introduced a new lower bound for the full-feedback setting (Theorem 3) and improved the presentation in several areas, for example by highlighting that other works adopt similar assumptions and by adding Figure 1. While the questions raised by the Reviewer are surely interesting, addressing them would require a complete rethinking of our work, as also pointed out by the Reviewer. In summary:
>
> - Within our framework, it is provably not possible to be persuasive in the first rounds, since no information about the environment is available. This challenge has been addressed by other works in similar settings (see, e.g., [Bernasconi et al., 2022, Cacciamani et al., 2023, Gan et al., 2023]) by considering the total violation of persuasiveness, and thus we believe that this is a reasonable approach that deserves to be investigated in MPPs as well.
>
> - Addressing the case in which the receivers are no-regret learners would require completely different approaches, outside the scope of our paper. Nevertheless, we believe that following this research direction would surely be interesting in future works.
>
> - Finally, let us remark that, without sender’s knowledge of receivers’ utilities, introducing robustification/pessimism into signaling schemes is of no help, unless one completely changes our model by introducing additional assumptions.
>
> We are committed to expanding the introduction in the final version of the paper to better clarify the limitations raised by the Reviewer.

---

> > ### Comment · Reviewer_ocM5 · 2025-08-03
> >
> > I am satisfied with responses and would like keep my positive rating.

---

### Official Review · Reviewer_kwa7 · 2025-07-05

**Clarity:** 4
**Significance:** 3
**Originality:** 3
**Rating:** 5
**Confidence:** 4

**Summary:**

This paper considers the problem of solving Markov Bayesian persuasion problems when the sender knows nothing about the game and proposes a learning algorithm that finds the solution with sublinear regret. Moreover, the authors show that the proposed bound is tight.

**Questions:**

Can the framework be extended to the case where the learning steps are limited, or where each step incurs a cost?

**Ethical Concerns:**

["NO or VERY MINOR ethics concerns only"]

**Final Justification:**

I think the paper provides solid results, and the author response has answered my question in a satisfactory way.

**Limitations:**

yes

**Paper Formatting Concerns:**

I did not find any formatting issue in the paper.

**Quality:**

3

**Strengths And Weaknesses:**

The problem studied in the paper is interesting and practical. No assumption is made about the sender's knowledge about the game. This significantly improve the applicability of the Markov Bayesian persuasion model. The paper is well-written and easy to follow. The proposed method is similar to the bandit algorithms where optimistic actions are taken. I like the idea of the exploration phase in the partial feedback case. Such a phase provides a good enough initial guess but affects the overall regret. The regret can be tuned by controlling the length of the phase. The tightness result is strong, though not surprising.

Overall, I think this is a solid theory paper. The results could also open up new research opportunities for similar problems.

---

> ### Author Rebuttal · Authors · 2025-07-30
>
> We thank the Reviewer for the positive evaluation of our work.
>
> Re: *"Can the framework be extended to the case where the learning steps are limited, or where each step incurs a cost?"*
>
> For the limited steps case, our algorithm needs to know the time horizon in advance. Differently, the case where the learner incurs an additional cost during the learning dynamic can be addressed by employing standard techniques for constrained Markov decision processes (see, e.g., [1]), assuming that the cost depends on the state-action pair only.
>
> [1] "Constrained Markov decision processes", Altman (1999)

---

> > ### Comment · Reviewer_kwa7 · 2025-08-06
> >
> > Thank you for the response. That aligns with my understanding of the problem.

---

### Official Review · Reviewer_JFBd · 2025-07-06

**Clarity:** 4
**Significance:** 3
**Originality:** 4
**Rating:** 5
**Confidence:** 4

**Summary:**

The paper gives an interesting extension of the so called MPP (Markov Persuasion Process) by introducing the learning component. The paper considers the setting of MPP where both senders and receivers have no knowledge of the (implicit) "markovian" environment's transitions and rewards. The paper addresses non-trivial extensions to MPP with a novel solution (specially relating the occupancy measure to the MPP formulation).

**Questions:**

see weaknesses

**Ethical Concerns:**

["NO or VERY MINOR ethics concerns only"]

**Final Justification:**

raised my score to 5
I found this paper interesting; perhaps it can lead to multiple research directions in the future.

**Limitations:**

Need discussions on: the practical utility of the MPP model; computational traceability of the proposed algorithms

**Quality:**

3

**Strengths And Weaknesses:**

S:

1) Extremely well written with a clear model description, especially the first three sections.

2) While the setting is still not practical in the current format, the MPP formulation definitely deserves a good consideration and analysis. Involving learning, this paper could give a reasonable first one.

3) Occupancy measure and signaling policy equivalence is very insightful and may be useful independently.
W:

1)I learned about MPP while reading this submission. I do not follow what do the states of the MDP represent for any practical situation. Can the authors explain that? Further, what are some practical examples where MPP might be useful?

2)Why is the assumption that a new receiver observes a_k at each time step important?

3)In their model, the sender/principal commits to a signaling policy throughout the play. This seems to be very restrictive. Very reasonably, I would assume the signal policy is allowed to be adopted as new information is revealed.

4)A minor comment: In RL terminology, partial feedback (as referred to by the authors) is called bandit feedback in most other papers. While the full feedback (as referred to by the authors) is still weaker (as rightly pointed out in the footnote). My suggestion would be to to call the full feedback as simply partial feedback

5)In Lines 150-155, the authors argue for a direct and persuasive signal policy for a one-shot Bayesian persuasion framework. The argument is not clear. Further, the actual algorithm 2 does not play a persuasive policy. It is not clear what the structure of the sender's signals is with algorithm 2. What is the significance of playing a persuasive policy?

6)The MDP structure of MPP induces a POMDP - Partially observable MDP. The authors should include this characterization as well.

7)A lot of information is pushed to the appendix about Problem 2, as solved in Algorithm 2. A summary of the optimization program will be helpful. e.g., what does the objective look like, the main constraints? I am concerned about the computational traceability of the same. If it is tractable, what is the runtime? Further "optimistic exploration" is hidden in the objective of the optimization. Given the details of the main paper is hard to understand how the uncertainty quantification is being done by the use of optimism. Unfortunately, this is one of the biggest weaknesses of the paper.

---

> ### Author Rebuttal · Authors · 2025-07-30
>
> We thank the Reviewer for the positive evaluation of our work.
>
> Re: *I learned about MPP while reading this submission. I do not follow what do the states of the MDP represent for any practical situation. Can the authors explain that? Further, what are some practical examples where MPP might be useful?*
>
> In the following, we provide a simple illustrative example. Consider a product recommendation platform that suggests items to multiple users. In this setting, the platform acts as the sender, while the users are the receivers who select actions, such as clicking on or purchasing recommended products. The model distinguishes between two types of information: the outcome  ($\omega$), which represents the sender’s private knowledge (e.g., product features and availability), and the state ($s$), which captures the dynamic environment (e.g., user browsing history). Outcomes are drawn from a prior distribution and are used by the platform to personalize its suggestions. The state evolves over time based on the user’s action, the previous state, and the realized outcome. Both the sender’s and the receivers’ utilities depend on the chosen action, the current state, and the realized outcome.
>
> We will surely include this discussion in the final version of the paper.
>
> Re: *"Why is the assumption that a new receiver observes $a_k$ at each time step important?"*
>
> We thank the Reviewer for pointing out this aspect. In Bayesian persuasion problems, it is standard for the receiver to observe the action $a_k$, as this avoids tie-breaking issues among multiple optimal actions. (see, e.g., Kamenica and Gentzkow [2011]).
>
> Re: *In their model, the sender/principal commits to a signaling policy throughout the play. This seems to be very restrictive. Very reasonably, I would assume the signal policy is allowed to be adopted as new information is revealed.*
>
> We thank the Reviewer for the question. We remark that, in episodic loop-free MDPs---and similarly in episodic loop-free MPPs---updating the policy within an episode does not provide any benefit. This is because the information acquired during the early steps of an episode cannot be leveraged to improve predictions for the later steps within the same episode. Instead, such information only becomes useful in the first steps of the next episode.
>
> Re: *A minor comment: In RL terminology, partial feedback (as referred to by the authors) is called bandit feedback in most other papers. While the full feedback (as referred to by the authors) is still weaker (as rightly pointed out in the footnote). My suggestion would be to to call the full feedback as simply partial feedback*
>
> We thank the Reviewer for the observation. We will better specify these aspects in the final version of the paper.
>
> Re: *In Lines 150-155, the authors argue for a direct and persuasive signal policy for a one-shot Bayesian persuasion framework. The argument is not clear. Further, the actual algorithm 2 does not play a persuasive policy. It is not clear what the structure of the sender's signals is with algorithm 2. What is the significance of playing a persuasive policy?*
>
> In classical one-shot Bayesian persuasion, it can be shown that there is no loss of generality in restricting attention to signaling schemes in which the receiver is incentivized to follow the sender’s recommendation. These signaling schemes are called direct (the set of signals coincides with the set of actions) and persuasive (the receiver follows the recommended action).
>
> In our setting, we have no prior knowledge about the receiver at the beginning of the interaction, so we must slightly relax the persuasiveness constraint. Since this relaxation leads to a broader set of signaling schemes, it remains possible to achieve the utility of an optimal persuasive and direct signaling scheme (see, e.g., Kamenica and Gentzkow [2011]).
>
> Re: *The MDP structure of MPP induces a POMDP - Partially observable MDP. The authors should include this characterization as well.*
>
> The Reviewer is correct: our setup introduces partial observability of the (persuasiveness) cost incurred by the sender. We will clarify this point more explicitly in the final version of the paper.
>
> Re: *A lot of information is pushed to the appendix about Problem 2, as solved in Algorithm 2. A summary of the optimization program will be helpful. e.g., what does the objective look like, the main constraints? I am concerned about the computational traceability of the same. If it is tractable, what is the runtime? Further "optimistic exploration" is hidden in the objective of the optimization. Given the details of the main paper is hard to understand how the uncertainty quantification is being done by the use of optimism. Unfortunately, this is one of the biggest weaknesses of the paper."*
>
> Regarding Problem 2, we agree with the Reviewer that much of the information is deferred to the appendix. Unfortunately, this is due to the space constraints of the NeurIPS format. In the final version of the paper, we commit to inserting the LP in the main part, due to the extra page available.
>
> The optimization problem (Problem 2) solved at each round by our algorithm is a linear program with a number of variables and constraints polynomial in $|X|$, $|A|$, and $|\Omega|$. As such, it can be solved in time polynomial in these parameters. The actual runtime depends on the specific LP solver used but remains polynomial in the overall input size.

---

> > ### Comment · Reviewer_JFBd · 2025-08-05
> >
> > Thanks for the rebuttal.
> > I found this paper interesting; perhaps it can lead to multiple research directions in the future.

---

### Decision · Program_Chairs · 2025-09-17

**Decision:**

Accept (poster)

**Comment:**

The paper proposes a novel framework for Markov Persuasion Processes (MPPs) that incorporates a learning component, addressing the practical limitation of a sender having no prior knowledge of the environment's dynamics or receiver's preferences. The core contribution is the Optimistic Persuasive Policy Search (OPPS) algorithm, which achieves sublinear regret and persuasiveness violation guarantees. All reviewers were positive about the paper, acknowledging its well-written nature and the significant originality of the problem setting. The theoretical results, including the tight regret bounds and the insightful equivalence between occupancy measures and signaling policies, were highlighted as major strengths. The authors provided clear and satisfactory rebuttals to all questions, addressing concerns about the practical utility of the MPP model, the restrictiveness of a committed signaling policy, and the computational tractability of the optimization problems. The reviewers' final comments confirmed that their initial questions were fully resolved, with all maintaining or raising their scores to a positive recommendation for acceptance.